# BOUNDING CONDITIONAL VALUE-AT-RISK VIA AUXILIARY DISTRIBUTIONS WITH BOUNDED DISCREPANCIES

## ABSTRACT

In this paper, we develop a theoretical framework for bounding the CVaR of a random variable $X$ using another related random variable $Y$, under assumptions on their cumulative and density functions. Our results yield practical tools for approximating $\mathrm{CVaR}_\alpha(X)$ when direct information about $X$ is limited or sampling is computationally expensive, by exploiting a more tractable or observable random variable $Y$. Moreover, the derived bounds provide interpretable concentration inequalities that quantify how the tail risk of $X$ can be controlled via $Y$.

## 1 INTRODUCTION

Conditional Value-at-Risk (CVaR) has emerged as a critical risk measure in modern artificial intelligence, particularly in settings where decision-making under uncertainty and rare but severe events must be rigorously addressed. Unlike traditional expected value objectives, which can overlook tail risks, CVaR explicitly quantifies the expected loss in the worst-case fraction of scenarios, providing a principled framework for robust and risk-sensitive learning and planning. This property makes CVaR especially valuable in reinforcement learning, safe control, and robust optimization, where agents must balance performance with safety guarantees in the presence of model misspecification, adversarial disturbances, or highly uncertain environments. As AI systems increasingly operate in high-stakes domains—from autonomous driving to financial trading—the ability to reason about and control tail risks via CVaR is becoming an essential component of reliable and trustworthy AI.

The Conditional Value at Risk (CVaR) Rockafellar et al. (2000) is a widely used risk measure that facilitates the optimization of the upper tail of the cost's distribution. The dual representation of CVaR Artzner et al. (1999) enables its interpretation as the worst-case expectation of the cost Chow et al. (2015), thereby motivating its use for risk-averse decision making. CVaR is also a coherent risk measure with desirable properties for safe planning Majumdar & Pavone (2020), and its estimators have performance guarantees that ensure their reliability in practice Brown (2007); Thomas & Learned-Miller (2019).

In this paper, we derive bounds on the CVaR of a random variable $X$, given another random variable $Y$, where the relationship between $X$ and $Y$ is characterized in terms of their cumulative distribution functions (CDFs) and probability density functions (PDFs). These bounds have two key contributions in practice. First, they enable the approximation of $\mathrm{CVaR}_\alpha(X)$ when direct access to $X$ is limited, by leveraging a related distribution $Y$ for which information is accessible. Second, they lead to interpretable concentration inequalities, encompassing existing concentration bounds for $\mathrm{CVaR}_\alpha(X)$ as a special case. Specifically, they characterize the CVaR of $X$ through the CVaR of $Y$ with an appropriately adjusted confidence level.

Owing to space constraints, proofs that do not appear in the main text are provided in the Appendix, and the simulations code is available in the supplemental material. While the present work is mostly theoretical, it provides mathematical background needed for risk-averse online agents' deployment acceleration in reinforcement learning; a detailed motivation is given in Appendix A.

## 2 PRELIMINARIES

Let $X$ be a random variable defined on a probability space $(\Omega, \mathcal{F}, P)$, where $\mathcal{F} = 2^{\Omega}$ is the $\sigma$-algebra, and $P : \mathcal{F} \to [0, 1]$ is a probability measure. Assume further that $E|X| < \infty$. We denote the cumulative density function (CDF) of the random variable $X$ by $F_X(x) = P(X \leq x)$. The value at risk at confidence level $\alpha \in (0, 1)$ is the $1 - \alpha$ quantile of $X$, i.e

$$VaR_\alpha(X) \triangleq \inf\{x \in \mathbb{R} : F(x) > 1 - \alpha\} \tag{1}$$

For simplicity we denote $q_\alpha^X = VaR_\alpha(X)$. The conditional value at risk (CVaR) at confidence level $\alpha$ is defined as Rockafellar et al. (2000)

$$CVaR_\alpha(X) \triangleq \inf_{w \in \mathbb{R}} \{w + \frac{1}{\alpha}\mathbb{E}[(X - w)^+] | w \in \mathbb{R}\}, \tag{2}$$

where $(x)^+ = \max(x, 0)$. For a smooth $F$, it holds that Pflug (2000)

$$CVaR_\alpha(X) = \mathbb{E}[X | X > VaR_\alpha(X)] = \frac{1}{\alpha}\int_{1-\alpha}^{1} F^{-1}(v)dv. \tag{3}$$

Let $X_i \overset{\text{iid}}{\sim} F$ for $i \in \{1, \ldots, n\}$. Denote by

$$\hat{C}_\alpha(X) \triangleq \hat{C}_\alpha(\{X_i\}_{i=1}^n) \triangleq \inf_{x \in \mathbb{R}} \left\{ x + \frac{1}{n\alpha}\sum_{i=1}^{n}(X_i - x)^+ \right\} \tag{4}$$

the estimate of $CVaR_\alpha(X)$ Brown (2007). Theorem 2.1, that bounds the deviation of the estimated CVaR and the true CVaR, was proved in Brown (2007).

**Theorem 2.1** *If $supp(X) \subseteq [a, b]$ and $X$ has a continuous distribution function, then for any $\delta \in (0, 1]$,*

$$P\left(CVaR_\alpha(X) - \hat{C}_\alpha(X) > (b - a)\sqrt{\frac{5ln(3/\delta)}{\alpha n}}\right) \leq \delta, \tag{5}$$

$$P\left(CVaR_\alpha(X) - \hat{C}_\alpha(X) < -\frac{(b-a)}{\alpha}\sqrt{\frac{ln(1/\delta)}{2n}}\right) \leq \delta. \tag{6}$$

equation 4 can be expressed as

$$\hat{C}_\alpha(X) = X^{(n)} - \frac{1}{\alpha}\sum_{i=1}^{n-1}(X^{(i)} - X^{(i-1)})\left(\frac{i}{n} - (1 - \alpha)\right)^+, \tag{7}$$

where $X^{(i)}$ is the $i$th order statistic of $X_1, \ldots, X_n$ in ascending order Thomas & Learned-Miller (2019). The results presented in Theorems 2.2 and 2.3, following the work of Thomas & Learned-Miller (2019), yield tighter bounds on the CVaR compared to those established by Brown (2007).

**Theorem 2.2** *If $X_1, \ldots, X_n$ are independent and identically distributed random variables and $\Pr(X_1 \leq b) = 1$ for some finite $b$, then for any $\delta \in (0, 0.5]$,*

$$\Pr\left(\text{CVaR}_\alpha(X_1) \leq Z_{n+1} - \frac{1}{\alpha}\sum_{i=1}^{n}(Z_{i+1} - Z_i) \times \left(\frac{i}{n} - \sqrt{\frac{ln(1/\delta)}{2n}} - (1 - \alpha)\right)^+\right) \geq 1 - \delta, \tag{8}$$

*where $Z_1, \ldots, Z_n$ are the order statistics (i.e., $X_1, \ldots, X_n$ sorted in ascending order), $Z_{n+1} = b$, and $x^+ \triangleq \max\{0, x\}$ for all $x \in \mathbb{R}$.*

**Theorem 2.3** *If $X_1, \ldots, X_n$ are independent and identically distributed random variables and $\Pr(X_1 \geq a) = 1$ for some finite $a$, then for any $\delta \in (0, 0.5]$,*

$$\Pr\left(\text{CVaR}_\alpha(X_1) \geq Z_n - \frac{1}{\alpha}\sum_{i=0}^{n-1}(Z_{i+1} - Z_i) \times \left(\min\left\{1, \frac{i}{n} + \sqrt{\frac{ln(1/\delta)}{2n}}\right\} - (1 - \alpha)\right)^+\right)$$
$$\geq 1 - \delta, \tag{9}$$

*where $Z_1, \ldots, Z_n$ are the order statistics (i.e., $X_1, \ldots, X_n$ sorted in ascending order), $Z_0 = a$, and $x^+ \triangleq \max\{0, x\}$ for all $x \in \mathbb{R}$.*

## 3 PROBLEM FORMULATION

Let $X$ and $Y$ be two random variables. This paper seeks to establish upper and lower bounding functions, $f^U$ and $f^L$, which take as input the random variable $Y$ and a confidence level $\alpha$, and satisfy

$$f^L(Y, \alpha) \leq CVaR_\alpha(X) \leq f^U(Y, \alpha). \tag{10}$$

For instance, one may take $X$ to be a Gaussian mixture distribution and $Y$ a normal distribution, in which case the bound provides a control of the Gaussian mixture's CVaR in terms of the normal distribution.

## 4 THEORETICAL CVAR BOUNDS

In this section, we establish bounds for the CVaR of the random variable $X$ by leveraging an auxiliary random variable $Y$. Two forms of distributional relationships between their respective CDFs are considered: a uniform bound and a non-uniform bound, as illustrated in Figure 1b and Figure 1c. These results provide the theoretical basis for subsequent sections, in which the derived bounds are applied to estimate the CVaR of random variables that are either computationally intractable or prohibitively expensive to sample directly.

Theorem 4.1 bounds $\text{CVaR}_\alpha(X)$ in terms of $\text{CVaR}_\alpha(Y)$ under the sole condition that the cumulative distribution functions of $X$ and $Y$ differ by at most a known uniform bound. This representation enhances the interpretability of the bound and constitutes a novel aspect of the result, made possible by framing the bounding problem in terms of distributional discrepancies. In the next section, we demonstrate that the bounds established in Thomas & Learned-Miller (2019) (theorems 2.2 and 2.3) arise as a special case of Theorem 4.1, thereby providing an interpretation for existing CVaR bounds that are otherwise difficult to interpret.

**Theorem 4.1** *Let $X$ and $Y$ be random variables and $\epsilon \in [0, 1]$.*

1. **Upper bound:** *assume that $P(X \leq b_X) = 1, P(Y \leq b_Y) = 1$ and $\forall z \in \mathbb{R}, F_Y(z) - F_X(z) \leq \epsilon$, then*

   (a) *If $\alpha > \epsilon$,*

   $$CVaR_\alpha(X) \leq \frac{\epsilon}{\alpha}max(b_X, b_Y) + (1 - \frac{\epsilon}{\alpha})CVaR_{\alpha-\epsilon}(Y). \tag{11}$$

   (b) *If $\alpha \leq \epsilon$, then $CVaR_\alpha(X) \leq max(b_X, b_Y)$.*

2. **Lower Bound:** *If $P(X \geq a_X) = 1, P(Y \geq a_Y) = 1$ and $\forall z \in \mathbb{R}, F_X(z) - F_Y(z) \leq \epsilon$, then*

   (a) *If $\alpha + \epsilon \leq 1$,*

   $$CVaR_\alpha(X) \geq (1 + \frac{\epsilon}{\alpha})CVaR_{\alpha+\epsilon}(Y) - \frac{\epsilon}{\alpha}CVaR_\epsilon(Y) \tag{12}$$

   (b) *If $\alpha + \epsilon > 1$ and $a_{min} = min(a_X, a_Y)$,*

   $$CVaR_\alpha(X) \geq \mathbb{E}[Y] - \epsilon CVaR_\epsilon(Y) + (\alpha + \epsilon - 1)a_{min} \tag{13}$$

*Proof.* Due to space constraints, we provide only a proof sketch here; the complete proof can be found in the supplementary material.

The strategy of the proof is to construct two distributions derived from $F_Y$, denoted $F_Y^L$ and $F_Y^U$, such that $F_X$ is first-order stochastically dominated by $F_Y^U$ and first-order stochastically dominates $F_Y^L$; that is, $\forall x \in \mathbb{R}, F_Y^L(x) \geq F_X \geq F_Y^U(x)$. Consequently, since CVaR is a coherent risk measure, it follows that $CVaR_\alpha^{F_Y^L} \leq CVaR_\alpha^{F_X} \leq CVaR_\alpha^{F_Y^U}$, where $CVaR_\alpha^{F_Y^L}$, $CVaR_\alpha^{F_X}$, and $CVaR_\alpha^{F_Y^U}$ denote the CVaR at level $\alpha$ corresponding to the distributions $F_Y^L$, $F_X$, and $F_Y^U$, respectively.

Let $a_{\min} = \min(a_X, a_Y)$ and $b_{\min} = \min(b_X, b_Y)$, and assume $P(a_X \leq X \leq b_X) = 1, P(a_Y \leq Y \leq b_Y) = 1$. Define the interval $[b_{\min}, b_X]$ to be $\emptyset$ if $b_{\min} = b_X$, and equal to $[b_Y, b_X]$ otherwise.

Analogously, define $[a_{\min}, a_X)$ in the same manner. We then define upper and lower bounds for $F_Y$ as follows (see Figure 1a):

$$F_Y^U(y) = \begin{cases} 0 & y < \max(a_X, q_{1-\epsilon}^Y) \\ \min(F_Y(y) - \epsilon, 1 - \epsilon) & y \in [\max(a_X, q_{1-\epsilon}^Y), \\ & \quad , b_{\min}) \\ 1 - \epsilon & y \in [b_{\min}, b_X) \\ 1 & y \geq \max(b_X, b_Y) \end{cases} \tag{14}$$

$$F_Y^L(y) = \begin{cases} 0 & y < a_{min} \\ \epsilon & y \in [a_{min}, a_Y) \\ min(F_Y(y) + \epsilon, 1) & y \in [a_Y, min(q_\epsilon^Y, b_X)) \\ 1 & y \geq min(q_\epsilon^Y, b_X). \end{cases} \tag{15}$$

As a first step, it is necessary to verify that $F_Y^L$ and $F_Y^U$ are valid cumulative distribution functions and that they satisfy $F_Y^L \leq F_X \leq F_Y^U$. This verification is deferred to the supplementary material. Figure 1a illustrates the corresponding upper and lower bounding CDFs.

**Upper bound for $CVaR_\alpha^{F_Y^U}$:** From Acerbi & Tasche (2002), CVaR is equal to an integral of the VaR

$$CVaR_\alpha^{F_Y^U} = \frac{1}{\alpha} \int_{1-\alpha}^1 \inf\{y \in \mathbb{R} : F_Y^U(y) \geq \tau\} d\tau = \frac{1}{\alpha} \int_{1-\alpha+\epsilon}^{1+\epsilon} \inf\{y \in \mathbb{R} : F_Y^U(y) \geq \tau - \epsilon\} d\tau \tag{16}$$

If $\alpha \leq \epsilon$, then $1 - \alpha + \epsilon \geq 1$, rendering the bound in the preceding equation trivial, as it attains the maximum value of the support of both $X$ and $Y$.

$$\frac{1}{\alpha} \int_{1-\alpha+\epsilon}^{1+\epsilon} \inf\{y \in \mathbb{R} : F_Y^U(y) \geq \tau - \epsilon\} \leq \frac{1}{\alpha} \int_{1-\alpha+\epsilon}^{1+\epsilon} max(b_X, b_Y) d\tau = max(b_X, b_Y). \tag{17}$$

That is, $CVaR_\alpha(X) \leq max(b_X, b_Y)$.

If $\alpha > \epsilon$, then $1 - \alpha + \epsilon < 0$, and the integral may be decomposed into one term that is trivially bounded by the maximum of the support and another term that can be computed explicitly

$$CVaR_\alpha^{F_Y^U} = \frac{1}{\alpha} \int_{1-\alpha+\epsilon}^{1+\epsilon} \inf\{y \in \mathbb{R} : F_Y^U(y) \geq \tau - \epsilon\} d\tau$$

$$= \frac{1}{\alpha} [\underbrace{\int_1^{1+\epsilon} \inf\{y \in \mathbb{R} : F_Y^U(y) \geq \tau - \epsilon\} d\tau}_{\triangleq A_1} + \underbrace{\int_{1-\alpha+\epsilon}^1 \inf\{y \in \mathbb{R} : F_Y^U(y) \geq \tau - \epsilon\} d\tau}_{\triangleq A_2}] \tag{18}$$

The term $A_1$ is bounded, in a manner analogous to equation 45, by $\epsilon \max(b_X, b_Y)$, which is essentially the tightest bound attainable given the definition of $F_Y^U$. The term $A_2$ can be expressed in terms of the CVaR of $Y$, evaluated at a shifted confidence level. Specifically, the variable $\tau - \epsilon$ in $A_2$ lies within the interval $[1 - \alpha, 1 - \epsilon]$. Over this range, for all $y \in [\max(a_X, q_{1-\epsilon}^Y), b_{\min}]$, the inequality $F_Y^U(y) \leq F_Y(y) - \epsilon$ holds. This follows because if $q_{1-\epsilon}^Y \geq a_X$, then by definition $F_Y^U(y) = min(F_Y(y) - \epsilon, 1 - \epsilon)$, and if $a_X > q_{1-\epsilon}^Y$, then $F_Y^U(y) = 0 \leq F_Y(y) - \epsilon$ for $y \in [q_{1-\epsilon}^Y, a_X]$ and again $F_Y^U(y) = min(F_Y(y) - \epsilon, 1 - \epsilon)$ for $y \in [a_X, b_{min}]$.

$$A_2 \leq \int_{1-\alpha+\epsilon}^1 \inf\{y \in \mathbb{R} : F_Y(y) - \epsilon \geq \tau - \epsilon\} d\tau = \int_{1-(\alpha-\epsilon)}^1 \inf\{y \in \mathbb{R} : F_Y(y) \geq \tau\} d\tau \tag{19}$$

$$= (\alpha - \epsilon) CVaR_{\alpha-\epsilon}(Y).$$

Note that the confidence level is valid, as $\alpha, \epsilon \in (0, 1)$ and $\alpha > \epsilon$ imply $\alpha - \epsilon \in (0, 1)$. By combining the previous two expressions, we obtain a single bound:

$$CVaR_\alpha^{F_Y^U} \leq \epsilon \frac{max(b_X, b_Y)}{\alpha} + (1 - \frac{\epsilon}{\alpha}) CVaR_{\epsilon-\alpha}(Y). \tag{20}$$

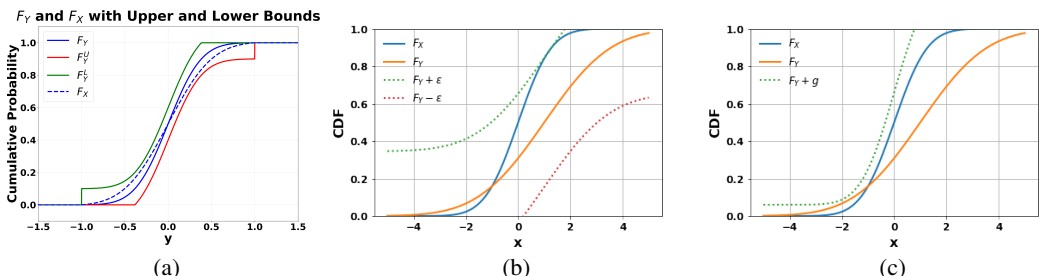

Figure 1: Illustrations of bounds on $F_X(x)$. (a) Bounds on $F_X$. (b) Bounds from Theorem 4.1. (c) Bounds from Theorem 4.3, where $g$ depends on $x$ and yields a tighter result than $F_Y(x) + \epsilon$.

In the case where $a_X = -\infty$, we have $\max(a_X, q_{1-\epsilon}^Y) = q_{1-\epsilon}^Y$, and the definition of $F_Y^U$ no longer involves $a_X$. Therefore, the inequality remains valid in the case of $a_X = -\infty$.

**Lower bound for** $CVaR_\alpha^{F_Y^L}$**:** The detailed proof is provided in the supplementary material. The proof follows an approach analogous to that of the upper bound: specifically, the integral defining $CVaR_\alpha^{F_Y^L}$ is decomposed into terms that exhibit the structure of a CVaR expression. $\qquad\square$

The parameter $\epsilon$ quantifies the discrepancy between the random variables $X$ and $Y$, and is defined as a uniform bound on the difference between their cumulative distribution functions. By construction, $\epsilon$ takes values in the interval $[0, 1]$. In the limiting case where $\epsilon = 1$, $Y$ provides no information about $X$. In this setting, the bounds provided by Theorem 4.1 reduce to trivial bounds on $\text{CVaR}_\alpha(X)$ involving the essential supremum of both $X$ and $Y$, rendering them ineffective for practical use. When $\epsilon = 0$, the cumulative distribution functions of $X$ and $Y$ are identical, and thus the bound becomes exact, yielding $\text{CVaR}_\alpha(X) = \text{CVaR}_\alpha(Y)$.

**Theorem 4.2** *Under the definitions of $X$, $Y$, $\epsilon$, and $\alpha$ specified in Theorem 4.1, the lower and upper bounds established therein converge to $\text{CVaR}_\alpha(X)$ as $\epsilon \to 0$.*

In the non-extreme cases, it holds that $\alpha > \epsilon$ and $\alpha + \epsilon \le 1$, reducing the bounding problem to equation 69 and equation 70. As we will see in the next section, $\epsilon$ could be very close to zero in practice. For $\alpha > \epsilon$, the lower bound on $\text{CVaR}_\alpha(X)$ is a weighted average of the CVaR of $Y$ at a confidence level shifted by the distributional discrepancy $\epsilon$ and the maximum of the supports of $X$ and $Y$, where the weights are proportional to the amount of distributional discrepancy. For $\alpha + \epsilon \le 1$, by restructuring equation 70 as follows,

$$CVaR_\alpha(X) \ge CVaR_{\alpha+\epsilon}(Y) + \frac{\epsilon}{\alpha}(CVaR_{\alpha+\epsilon}(Y) - CVaR_\epsilon(Y)). \tag{21}$$

The lower bound corresponds to the CVaR of $Y$ evaluated at a confidence level adjusted by the distributional discrepancy $\epsilon$, augmented by a correction term that is proportional to the distributional discrepancy.

Theorem 4.1 assumes that the parameter $\epsilon$ provides an upper bound on the pointwise difference between the cumulative distribution functions $F_X$ and $F_Y$. As illustrated in Figure 1b, this bound is particularly conservative in the vicinity of $x = -1$, where the actual discrepancy between $F_X$ and $F_Y$ is significantly smaller than the global bound $\epsilon$. Ideally, a tighter bound on $F_X(x)$ would allow for variation with respect to $x$, rather than relying on a uniform constant. Specifically, one seeks a pointwise bound of the form $F_X(x) \le F_Y(x) + g(x)$ for some non-negative function $g$, as illustrated in Figure 1c. In this paper, we defer the study of specific choices of the function $g$ to future work, and instead provide general assumptions under which a feasible tighter bound can be established using such a function.

**Theorem 4.3** *(Tighter CVaR Lower Bound) Let $\alpha \in (0, 1)$, $X$ and $Y$ be random variables. Define the a random variable $Y^L$ such that $F_{Y^L}(y) \triangleq \min(1, F_Y(y) + g(y))$ for $g : \mathbb{R} \to [0, \infty)$. Assume $\lim_{x \to -\infty} g(x) = 0$, $g$ is continuous from the right and monotonic increasing. If $\forall x \in \mathbb{R}$, $F_X(x) \le F_Y(x) + g(x)$, then $F_{Y^L}$ is a CDF and $CVaR_\alpha(Y^L) \le CVaR_\alpha(X)$.*

Theorem 4.3 defines a random variable $Y^L$, constructed from $Y$ and the function $g$, such that the distributional discrepancy between $Y^L$ and $X$ is determined explicitly by $g$ rather than being uniformly bounded as in Theorem 4.1. In addition, it offers a criterion for determining whether a given function $g$ can be used to derive a lower bound on the CVaR. This bound extends the lower bound established in Theorem 4.1, which is obtained when $g(x)$ is constant and equal to $\epsilon$ for all $x \in \mathbb{R}$.

Note that in Theorem 4.3, the function $g$ is assumed to be non-decreasing and right-continuous. If one assumes only that $|F_X(x) - F_Y(x)| \leq g(x)$ for some function $g : \mathbb{R} \to [0, \infty)$, which is not necessarily monotonic or continuous, then the most general form of the CVaR bounds is given by

$$CVaR_\alpha(X) = \frac{1}{\alpha} \int_{1-\alpha}^{1} \inf\{z \in \mathbb{R} : F_X(z) \geq \tau\} d\tau \geq \frac{1}{\alpha} \int_{1-\alpha}^{1} \inf\{z \in \mathbb{R} : F_Y(z) + g(z) \geq \tau\} d\tau,$$
(22)

$$CVaR_\alpha(X) = \frac{1}{\alpha} \int_{1-\alpha}^{1} \inf\{z \in \mathbb{R} : F_X(z) \geq \tau\} d\tau \leq \frac{1}{\alpha} \int_{1-\alpha}^{1} \inf\{z \in \mathbb{R} : F_Y(z) - g(z) \geq \tau\} d\tau.$$
(23)

Another option is to specify the distributional discrepancy through the density functions underlying the cumulative distribution functions. Let $f_x$ and $f_y$ be the probability density functions of $X$ and $Y$, respectively, and let $h : \mathbb{R} \to [0, \infty)$ describe the pointwise discrepancy between them. Theorem 4.4 specifies conditions on the function $h$ under which a lower bound for the CVaR of $X$ can be obtained. A key advantage is that this bound takes the form of the CVaR of a random variable, enabling its estimation with performance guarantees via CVaR concentration bounds given in Theorems 2.1, 2.3, 2.2 and Theorem 5.1.

**Theorem 4.4** *Let $\alpha \in (0, 1)$, $X$ and $Y$ random variables. Define $h : \mathbb{R} \to [0, \infty)$ to be a continuous function, $g(z) \triangleq \int_{-\infty}^{z} h(x) dx$ and $Y^L$ to be a random variable such that $F_{Y^L}(y) \triangleq \min(1, F_Y(y) + g(y))$. If $\lim_{z \to -\infty} g(z) = 0$ and $\forall x \in \mathbb{R}, f_x(x) \leq f_y(z) + h(x)$, then $F_{Y^L}$ is a CDF and $CVaR_\alpha(Y^L) \leq CVaR_\alpha(X)$.*

Theorem 4.4 is obtained as a corollary of Theorem 4.3 by defining a function $g$, as illustrated in Figure 1c, in terms of the function $h$. This construction demonstrates that the function $g$ in Theorem 4.3 can be generated from a broad class of density discrepancy functions.

## 5 CONCENTRATION INEQUALITIES

In this section, we derive concentration inequalities for $\text{CVaR}_\alpha(X)$ based on samples drawn from an auxiliary random variable $Y$. A notable special case of these inequalities arises when $Y$ is taken to follow the empirical cumulative distribution function (ECDF) of $X$. In such cases, concentration inequalities for $\text{CVaR}_\alpha(X)$ are obtained.

Let $X_1, \ldots, X_n \overset{i.i.d.}{\sim} F_X$, where $F_X$ is the CDF of a random variable $X$. The ECDF based on these samples is defined by $\hat{F}_X(x) = \frac{1}{n} \sum_{i=1}^{n} 1_{X_i \leq x}$, for $x \in \mathbb{R}$. Let $E_{\hat{F}_X}[X]$ denote the expectation with respect to $\hat{F}_X$, and let $C_\alpha^{\hat{F}_X}$ denote the CVaR computed under the empirical distribution $\hat{F}_X$.

**Theorem 5.1** *Let $X$ a random variable, $\alpha \in (0, 1], \delta \in (0, 0.5), \epsilon = \sqrt{\ln(1/\delta)/(2n)}$. Let $X_1, \ldots, X_n \overset{iid}{\sim} F_X$ be random variables that define the ECDF $\hat{F}_X$.*

1. ***Upper Bound:*** *If $P(X \leq b) = 1$, then*

    (a) *If $\alpha > \epsilon$ then $P(CVaR_\alpha(X) \leq (1 - \frac{\epsilon}{\alpha}) C_{\alpha-\epsilon}^{\hat{F}_X} + \frac{\epsilon}{\alpha} b) > 1 - \delta$.*
    (b) *If $\alpha \leq \epsilon$ then $CVaR_\alpha(X) \leq b$*

2. ***Lower Bound:*** *If $P(X \geq a) = 1$, then*

    (a) *If $\alpha + \epsilon < 1$, then $P(CVaR_\alpha(X) \geq (1 + \frac{\epsilon}{\alpha}) C_{\alpha+\epsilon}^{\hat{F}_X} - \frac{\epsilon}{\alpha} C_\epsilon^{\hat{F}_X}) > 1 - \delta$.*
    (b) *If $\alpha + \epsilon \geq 1$, then $P(CVaR_\alpha(X) \geq \frac{1}{\alpha}[(\alpha + \epsilon - 1)a + E_{\hat{F}_X}[X] - \epsilon C_\epsilon^{\hat{F}_X}]) > 1 - \delta$.*

Theorem 5.1 provides concentration inequalities for $\text{CVaR}_\alpha(X)$ in a form that enables the user to specify a desired bound consistency level $\delta$, which determines the probability that the bound holds. This result follows as a corollary of Theorem 4.1, in which the auxiliary random variable $Y$ is instantiated as the ECDF of $X$, whereas $X$ denotes the underlying true random variable, which is inaccessible in practice. The distributional discrepancy required by Theorem 4.1, denoted by $\epsilon$, is controlled in Theorem 5.1 via the Dvoretzky–Kiefer–Wolfowitz (DKW) inequality Dvoretzky et al. (1956). The DKW inequality ensures that the supremum distance between the true CDF and the ECDF converges to zero at a rate of order $1/\sqrt{n}$ as the number of samples $n$ increases. Corollary 5.2 establishes the asymptotic convergence of the bounds given in Theorem 5.1 as the sample size tends to infinity.

**Corollary 5.2** *Let $X$ be a random variable, $\alpha \in (0,1], \delta \in (0,0.5), \epsilon = \sqrt{\ln(1/\delta)/(2n)}, a \in \mathbb{R}, b \in \mathbb{R}, \eta > 0$. Let $X_1, \ldots, X_n \overset{iid}{\sim} F_X$ be random variables that define the ECDF $\hat{F}_X$. Denote by $U(n)$ and $L(n)$ the upper and lower bounds respectively from Theorem 5.1, where $n$ is the number of samples, then*

1. *If $P(X \leq b) = 1$, then $\lim_{n \to \infty} U(n) \overset{a.s}{=} CVaR_\alpha(X)$.*

2. *If $P(X \geq a) = 1$, then $\lim_{n \to \infty} L(n) \overset{a.s}{=} CVaR_\alpha(X)$.*

*where a.s denotes almost sure convergence.*

The concentration bounds established by Thomas & Learned-Miller (2019) coincide with those given in Theorem 5.1, rendering the results of Thomas & Learned-Miller (2019) a special case of Theorem 5.1. This equivalence arises because both Theorem 5.1 and Thomas & Learned-Miller (2019) derive concentration bounds for $\text{CVaR}_\alpha(X)$ by constructing an alternative CDF that stochastically dominates the true distribution $F_X$, employing the DKW inequality to control the discrepancy. The principal distinction between the two results lies in the formulation of the CVaR bound: Thomas & Learned-Miller (2019) express the bound through a sum of reweighted order statistics (theorems 2.2 and 2.3), resulting in a more intricate form, whereas Theorem 5.1 presents a more interpretable bound in terms of CVaR. The interpretability of these bounds constitutes a contribution of this paper.

Theorem 5.2 provides concentration inequalities for the theoretical $CVaR_\alpha(X)$ based on samples drawn from an auxiliary random variable $Y$, assuming only a bound on the distributional discrepancy between $X$ and $Y$.

**Theorem 5.3** *Let $X$ and $Y$ be random variables, $\epsilon \in [0,1], \delta \in (0,0.5)$, and $\eta = \sqrt{\ln(1/\delta)/(2n)}, \epsilon' = min(\epsilon + \eta, 1)$. Let $Y_1, \ldots, Y_n$ be independent and identically distributed samples from $F_Y$, and denote by $\hat{F}_Y$ the associated empirical cumulative distribution function.*

1. ***Upper Bound:*** *If $\forall z \in \mathbb{R}, F_Y(z) - F_X(z) \leq \epsilon$ and $P(X \leq b_X) = 1, P(Y \leq b_Y) = 1$, then*

   (a) *If $\alpha > \epsilon'$ then $P\left(CVaR_\alpha(X) \leq \frac{\epsilon'}{\alpha} max(b_X, b_Y) + (1 - \frac{\epsilon'}{\alpha})CVaR_{\alpha-\epsilon'}^{\hat{F}_Y^L}\right) > 1 - \delta$.*

   (b) *If $\alpha \leq \epsilon'$, then $CVaR_\alpha(X) \leq max(b_X, b_Y)$*

2. ***Lower Bound:*** *If $\forall z \in \mathbb{R}, F_X(z) - F_Y(z) \leq \epsilon$ and $P(X \geq a_X) = 1, P(Y \geq a_Y) = 1$, then*

   (a) *If $\alpha + \epsilon' \leq 1$, then*

$$P\left(CVaR_\alpha(X) \geq (1 + \frac{\epsilon'}{\alpha})CVaR_{\alpha+\epsilon'}^{\hat{F}_Y^L} - \frac{\epsilon'}{\alpha}CVaR_{\epsilon'}^{\hat{F}_Y^L}\right) > 1 - \delta. \tag{24}$$

   (b) *If $\alpha + \epsilon' > 1$, then*

$$P\left(CVaR_\alpha(X) \geq \mathbb{E}_{\hat{F}_Y^L}[Y] - \epsilon'CVaR_{\epsilon'}^{\hat{F}_Y} + (\alpha + \epsilon' - 1)a_{min}\right) > 1 - \delta. \tag{25}$$

*Proof.* We begin by establishing that $\epsilon'$ bounds the distributional discrepancy between $\hat{F}_Y^L$ and $F_X$, under the assumption that $\sup_x \left( \hat{F}_Y^L(x) - F_Y(x) \right) \leq \epsilon$. Let $x \in \mathbb{R}$,

$$
\begin{aligned}
\hat{F}_Y^L(x) - F_X(x) &= \hat{F}_Y^L(x) - F_Y(x) + F_Y(x) - F_X(x) \\
&\leq |\hat{F}_Y^L(x) - F_Y(x)| + |F_Y(x) - F_X(x)| \leq \epsilon + \eta = \epsilon'.
\end{aligned}
\tag{26}
$$

Assume that $\alpha > \epsilon'$. Note that, conditional on the event $\sup_{z \in \mathbb{R}} \left( \hat{F}_Y(z) - F_X(z) \right) \leq \epsilon'$, the upper bound in Theorem 4.1 holds deterministically. Consequently, the probability that

$$
CVaR_\alpha(X) \leq \frac{\epsilon'}{\alpha} max(b_X, b_Y) + (1 - \frac{\epsilon'}{\alpha}) CVaR_{\alpha - \epsilon'}^{\hat{F}_Y^L}
\tag{27}
$$

holds is equal to one. From the law of total probability,

$$
P\left( CVaR_\alpha(X) \leq \frac{\epsilon'}{\alpha} max(b_X, b_Y) + (1 - \frac{\epsilon'}{\alpha}) CVaR_{\alpha - \epsilon'}^{\hat{F}_Y^L} \right) \geq P(\sup_{z \in \mathbb{R}} \left( \hat{F}_Y(z) - F_X(z) \right) \leq \epsilon').
\tag{28}
$$

The full derivation of the last inequality is available in the Appendix. From DKW Dvoretzky et al. (1956) inequality the following inequalities can be derived Thomas & Learned-Miller (2019)

$$
P\left( \sup_{x \in \mathbb{R}} \left( \hat{F}(x) - F(x) \right) \leq \sqrt{\frac{\ln(1/\delta)}{2n}} \right) \geq 1 - \delta, P\left( \sup_{x \in \mathbb{R}} \left( \hat{F}(x) - F(x) \right) \geq \sqrt{\frac{\ln(1/\delta)}{2n}} \right) \geq 1 - \delta.
\tag{29}
$$

$$
\begin{aligned}
P(\sup_{z \in \mathbb{R}} \left( \hat{F}_Y(z) - F_X(z) \right) \leq \epsilon') &\geq P(\sup_{z \in \mathbb{R}} \left( \hat{F}_Y(z) - F_Y(z) \right) + \sup_{z \in \mathbb{R}} \left( F_Y(z) - F_X(z) \right) \leq \epsilon') \\
&\geq P(\sup_{z \in \mathbb{R}} \left( \hat{F}_Y(z) - F_Y(z) \right) + \epsilon \leq \epsilon + \eta) = P(\sup_{z \in \mathbb{R}} \left( \hat{F}_Y(z) - F_Y(z) \right) \leq \eta) > 1 - \delta.
\end{aligned}
\tag{30}
$$

The first inequality follows from the triangle inequality; the second holds since $\epsilon$ bounds the distributional discrepancy between $X$ and $Y$; and the third follows from equation 76.

As with the preceding equations, all bounds in Theorem 4.1 hold deterministically for a given distributional discrepancy, where the discrepancy between $\hat{F}_Y$ and $F_X$ is $\epsilon'$. Consequently, the remaining probabilistic guarantees hold with probability at least $1 - \delta$. $\qquad\square$

The parameter $\epsilon$ in Theorem 5.2 captures the distributional discrepancy between $X$ and $Y$, consistent with its role in Theorem 4.1. Additionally, the theorem introduces $\epsilon'$ to represent the discrepancy between $F_X$ and $\hat{F}_Y$, where $\hat{F}_Y$ denotes the empirical CDF of $Y$ constructed from the sample $\{Y_i\}_{i=1}^n$. The parameter $\eta$ accounts for the additional distributional discrepancy beyond $\epsilon$, and captures the estimation error in approximating $F_Y$ using the empirical sample $\{Y_i\}_{i=1}^n$. By combining Theorem 4.1 with the DKW inequality Dvoretzky et al. (1956), we obtain probabilistic guarantees for the estimated bounds.

# 6 EXPERIMENTS

In this simulation study, we demonstrate how the CVaR of an inaccessible or computationally expensive distribution can be bounded using an auxiliary distribution that is more tractable. We then verify that the concentration bounds for $CVaR_\alpha(X)$ established in Theorem 5.1 coincide with those of Thomas & Learned-Miller (2019), which arise as a special case of our more general results. In our experiments, both approaches exhibit identical empirical performance. Specifically, we compare a truncated Gaussian Mixture Model (GMM) and its corresponding truncated Normal approximation for CVaR estimation under bounded support. Such a setting arises, for example, in POMDP planning, where a computationally expensive model used by the agent during online planning is replaced with a more tractable surrogate model, thereby improving the agent's decision-making speed Lev-Yehudi et al. (2024). In these settings, the agent's decisions may rely on bounds for the original value function that are derived from the tractable surrogate model Barenboim & Indelman (2022).

The GMM consists of five components with means $\mu_1 = 0.2$, $\mu_2 = -0.2$, $\mu_3 = -0.5$, $\mu_4 = 0.5$, $\mu_5 = 0.0$, variances $\sigma_1^2 = 0.5$, $\sigma_2^2 = 0.2$, $\sigma_3^2 = 0.1$, $\sigma_4^2 = 0.1$, $\sigma_5^2 = 0.3$, and weights $w_1 = 0.6$, $w_2 = 0.4$, $w_3 = 0.1$, $w_4 = 0.1$, $w_5 = 0.8$. The Normal approximation matches the GMM's mean and variance but cannot reproduce its multi-modal structure or outlier effects. All samples are

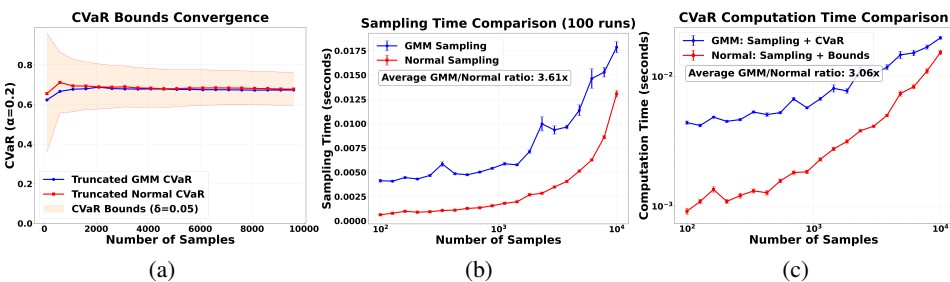

(a)          (b)          (c)

Figure 2: Figure 2a shows the bounds established in Theorem 5.3, demonstrating how a simple truncated Normal distribution can be employed to bound the CVaR of a truncated GMM. Figure 2b presents a comparison of the sampling times for each distribution, while Figure 2c reports the total computation time required to sample from the truncated GMM and estimate its CVaR, in contrast to sampling from the truncated Normal surrogate and computing the associated concentration bounds. All of the experiments were configured with confidence level $\alpha = 0.2$, and probability of error $\delta = 0.05$. Both Figures 2b and 2c display 95% confidence intervals around the empirical means.

truncated to the interval $[-1, 1]$, which reshapes the tails and slightly distorts boundary components. We compute CVaR at the 20% quantile for sample sizes ranging from 100 to 10,000, with 100 independent repetitions per setting to ensure statistical reliability. The distributional discrepancy ($\epsilon$ in Theorem 5.3) between the truncated GMM and the truncated Normal distribution is assessed via simulation, by estimating their respective cumulative distribution functions over a common set of bins and computing the maximum difference across all bins. This setup enables a direct assessment of the trade-off between computational efficiency and statistical accuracy when approximating a complex truncated mixture by a single truncated Normal distribution.

Figure 2b presents the sampling time comparison between the truncated GMM and the truncated Normal distribution, with an observed average time ratio of approximately 3.6 in favor of the Normal distribution. Figure 2c reports a total computational speedup of approximately 3.17 when comparing the process of sampling from the truncated GMM and estimating its CVaR to that of sampling from the truncated Normal and computing both upper and lower bounds as given in Theorem 5.3. Figure 2a illustrates the convergence behavior of the CVaR bounds as a function of the number of samples, comparing estimates obtained from the truncated GMM with bounds derived from the truncated Normal surrogate. It is important to note that these bounds are obtained without requiring full knowledge of the underlying GMM distribution. Instead, they rely solely on the discrepancy between the corresponding CDFs.

We evaluated the concentration bounds established by Thomas & Learned-Miller (2019) alongside our proposed bounds from Theorem 5.1 on a set of probability distributions: $\text{Beta}(2, 2)$, $\text{Beta}(0.5, 0.5)$, $\text{Beta}(2, 5)$, $\text{Beta}(5, 2)$, $\text{Beta}(10, 2)$, $\text{Beta}(2, 10)$, and the $\text{Laplace}(0, 1)$ distribution. These distributions were selected to match those used in the original study by Thomas & Learned-Miller (2019), enabling a direct comparison under identical conditions. For each distribution, our bounds precisely coincide with those reported by Thomas & Learned-Miller (2019), resulting in complete overlap between the two sets of bounds. The graphs exhibiting these results are available in the supplemental material.

## 7 CONCLUSIONS

This work presented interpretable concentration inequalities for $CVaR_\alpha(X)$, expressed in terms of the CVaR of an auxiliary random variable $Y$. The proposed bounds were validated through simulation studies, effectively bounding the CVaR of a truncated GMM by employing a less complex truncated Normal distribution. A comprehensive theoretical framework was developed for bounding CVaR using both uniform and non-uniform bounds on the cumulative and density functions of $X$ and $Y$. Notably, the state-of-the-art bounds introduced by Thomas & Learned-Miller (2019) emerge as a special case of the proposed general framework, which also offers a novel interpretation of their result.

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

APPENDIX

The Appendix is organized as follows. Section A provides motivation for the theoretical results. Section B contains the proofs of the theoretical bounds, while Section C presents the proofs of the concentration inequalities. Finally, Section D includes simulations demonstrating that our bounds are consistent with the results of Thomas & Learned-Miller (2019).

## A    MOTIVATION

This section aims to motivate the mathematical results presented in the paper. Although these results are general and not confined to any particular application, we illustrate their relevance using reinforcement learning. We begin with a brief overview of Partially Observable Markov Decision Processes (POMDPs), a generalization of Markov Decision Processes that constitutes the standard framework for reinforcement learning. Next, we define the reinforcement learning setting in terms of the value function and provide background on agent acceleration, commonly referred to as 'simplification' in the literature. Finally, we reinterpret the theoretical random variables $X$ and $Y$ from the main text in the context of reinforcement learning and demonstrate their significance in this setting.

### A.1    PARTIALLY OBSERVABLE MARKOV DECISION PROCESS

A finite-horizon Partially Observable Markov Decision Process (POMDP) is defined as the tuple $(X, A, Z, T, O, c, b_0)$, where $X$, $A$, and $Z$ denote the state, action, and observation spaces, respectively. The transition model $T(x_{t+1} \mid x_t, a_t) \triangleq P(x_{t+1} \mid x_t, a_t)$ specifies the probability of transitioning from state $x_t$ to $x_{t+1}$ given action $a_t$. The observation model is given by the conditional density $O(z_t \mid x_t) \triangleq P(z_t \mid x_t)$, representing the likelihood of observing $z_t$ given the underlying state $x_t$. Let $B$ denote the belief space (the set of all probability distributions over $X$), and define the stage-wise cost function as $c : B \times A \to \mathbb{R}$.

At each time step $t$, the agent maintains a belief $b_t \in B$, representing the posterior distribution over the state space given the history of actions and observations. The history is denoted by $H_t = \{z_{1:t}, a_{0:t-1}, b_0\}$, and the belief is defined as $b_t(x_t) \triangleq P(x_t \mid H_t)$ for $x_t \in X$.

A policy $\pi_t : B \to A$ at time $t$ is defined as a mapping from a belief state $b_t$ to an action $a_t = \pi_t(b_t)$. The state-dependent immediate cost incurred by executing action $a_t$ under belief $b_t$ is given by $c(b_t, a_t) \triangleq \mathbb{E}_{x \sim b_t}[c_x(x, a_t)]$, where $c_x(x, a_t)$ denotes the cost associated with taking action $a_t$ in state $x$, and is uniformly bounded by $R_{min} \leq |c_x(x, a_t)| \leq R_{\max}$. The cumulative cost over a finite horizon $T \in \mathbb{N}$, referred to as the *return*, is given by $R_{t:T} \triangleq \sum_{\tau=t}^{T} c(b_\tau, a_\tau)$, which serves as the performance criterion starting at time $t$.

The value function associated with a policy $\pi$ and initial belief $b_k$ is defined as

$$V^\pi(b_k) \triangleq \mathbb{E}[R_{k:T} \mid b_k, \pi] = \sum_{t=k}^{T} \mathbb{E}[c(b_t, a_t) \mid b_k, \pi], \tag{31}$$

and the corresponding $Q$-function is

$$Q^\pi(b_k, a_k) \triangleq \mathbb{E}_{z_{k+1}} \left[ c(b_k, a_k) + V^\pi(b_{k+1}) \mid b_k, a_k \right]. \tag{32}$$

A POMDP can equivalently be viewed as a belief-MDP, where the belief state $b_t$ serves as the fully observed state.

### A.2    SIMPLIFIED POMDP PLANNING

Simplification in POMDPs is employed to mitigate computational burden during online POMDP planning, as POMDPs are hard to solve Papadimitriou & Tsitsiklis (1987). The term simplification refers to a replacement of any component of a POMDP with a computationally cheaper alternative while providing formal performance guarantees on planning performance. Simplification of the observation model in POMDPs was studied in Lev-Yehudi et al. (2024), wherein the observation model was replaced with a computationally less expensive alternative, while deriving finite-sample convergence guarantees. **?** considered simplification of the state and observation spaces, provided deterministic guarantees.

As a formal example of observation model simplification, consider the original POMDP $M = (X, A, Z, T, O, c, b_0)$ and the simplified POMDP $M_s = (X, A, Z, T, O_s, c, b_0)$, which differ only in the observation model. In practice, the observation model $O$ may be represented by a computationally expensive neural network, whereas $O_s$ may correspond to a more lightweight neural network that is faster to sample from.

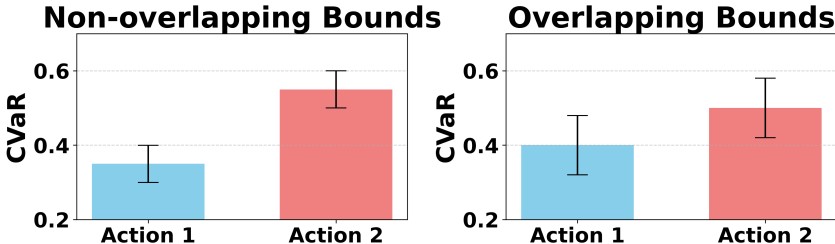

Figure 3: Illustration of action–sequence elimination based on bounding intervals. On the left, two actions are shown with disjoint bounds. In this case, the agent prefers action 1, as its associated action–value function is smaller. On the right, the two actions have overlapping bounds, and consequently the agent cannot distinguish between them based on the bounds.

One approach to accelerating agent decision making through model simplification is action elimination based on the simplified model. Let $V_M^\pi(b)$ and $Q_M^\pi(b)$ denote the value function and action-value function of the original POMDP $M$, and let $V_{M_s}^\pi(b)$ and $Q_{M_s}^\pi(b)$ denote the corresponding functions of the simplified POMDP $M_s$. A central aspect of such acceleration methods is the establishment of formal theoretical guarantees relating the simplified and original action-value functions, specifically by identifying conditions under which

$$P(|Q_M^\pi(b) - Q_{M_s}^\pi(b)| > \epsilon) \le \delta, \tag{33}$$

where $\epsilon > 0, \delta \in (0,1)$. In these cases, bounds for the original POMDP can be derived from the simplified POMDP, enabling the elimination of actions whose bounds do not overlap. An illustration of this procedure is given in Figure 3.

### A.3 CVaR-Based Risk-Averse RL

CVaR is a widely studied risk measure with numerous applications. In this example, we focus on one of its most direct formulations, also considered in Chow & Ghavamzadeh (2014), wherein the value function is defined as the CVaR of the return. Specifically,

$$V_M^{\pi,CVaR}(b_k, \alpha) \triangleq CVaR_\alpha[\sum_{t=k}^T c(b_t, \pi(b_t))|b_k, \pi], \tag{34}$$

$$Q_M^{\pi,CVaR}(b_k, a_k, \alpha) \triangleq CVaR_\alpha[c(b_k, a_k) + \sum_{t=k+1}^T c(b_t, \pi(b_t))|b_k, \pi]. \tag{35}$$

We denote by $V_{M_s}^{\pi,CVaR}$ and $Q_{M_s}^{\pi,CVaR}$ the value and action-value functions with respect to the simplified POMDP. This definition of the value function would be used in the next section.

### A.4 Connection To This Paper

In this section, we establish the connection between the random variables $X$ and $Y$ introduced earlier and the POMDP simplification framework. We first observe that, when the return is continuous with respect to the belief $b$, the properties of $CVaR$ imply that for $\alpha = 1$

$$V_M^{\pi,CVaR}(b, \alpha) = V_M^\pi(b), Q_M^{\pi,CVaR}(b, a, \alpha) = Q_M^\pi(b, a), \tag{36}$$

$$V_{M_s}^{\pi,CVaR}(b, \alpha) = V_{M_s}^\pi(b), Q_{M_s}^{\pi,CVaR}(b, a, \alpha) = Q_{M_s}^\pi(b, a). \tag{37}$$

Thus, the CVaR-based value function can be interpreted as a generalization of the standard expectation-based value function, since under mild conditions the latter arises as a special case of the former.

The POMDP determines the distribution of the return. Consequently, the results in this paper show that, if the distributional discrepancy between the return under the original POMDP and that under the simplified POMDP is bounded, one can bound the original value function in terms of the

simplified value function, thereby enabling accelerated decision making through the use of computationally cheaper models.

Formally, let $X$ and $Y$ in Theorem 5.3 correspond to the return distributions of the original POMDP $M$ and the simplified POMDP $M_s$, respectively. If the actual distributional discrepancy $\epsilon$ in Theorem 5.3 is known, the problem of accelerating agent decision making is effectively resolved, as existing results in the literature provide solutions for computing such bounds.

In the context of agent acceleration, the key application of this work is that, by providing bounds on the original value function of an MDP or POMDP in terms of a simplified model through the paper's results, the problem reduces to estimating the distributional discrepancy between the corresponding returns.

## B   PROOFS OF SECTION 4

**Theorem B.1** *(Theorem 4.1) Let $X$ and $Y$ be random variables and $\epsilon \in [0, 1]$.*

1. **Upper bound:** *assume that $P(X \leq b_X) = 1, P(Y \leq b_Y) = 1$ and $\forall z \in \mathbb{R}, F_Y(z) - F_X(z) \leq \epsilon$, then*

   (a) *If $\alpha > \epsilon$,*
   $$CVaR_\alpha(X) \leq \frac{\epsilon}{\alpha} max(b_X, b_Y) + (1 - \frac{\epsilon}{\alpha})CVaR_{\alpha-\epsilon}(Y). \tag{38}$$

   (b) *If $\alpha \leq \epsilon$, then $CVaR_\alpha(X) \leq max(b_X, b_Y)$.*

2. **Lower Bound:** *If $P(X \geq a_X) = 1, P(Y \geq a_Y) = 1$ and $\forall z \in \mathbb{R}, F_X(z) - F_Y(z) \leq \epsilon$, then*

   (a) *If $\alpha + \epsilon \leq 1$,*
   $$CVaR_\alpha(X) \geq (1 + \frac{\epsilon}{\alpha})CVaR_{\alpha+\epsilon}(Y) - \frac{\epsilon}{\alpha}CVaR_\epsilon(Y) \tag{39}$$

   (b) *If $\alpha + \epsilon > 1$ and $a_{min} = min(a_X, a_Y)$,*
   $$CVaR_\alpha(X) \geq \mathbb{E}[Y] - \epsilon CVaR_\epsilon(Y) + (\alpha + \epsilon - 1)a_{min} \tag{40}$$

*Proof.* The strategy of the proof is to construct two distributions derived from $F_Y$, denoted $F_Y^L$ and $F_Y^U$, such that $F_X$ is stochastically bounded between them; that is, $F_Y^L \leq F_X \leq F_Y^U$. Consequently, since CVaR is a coherent risk measure, it follows that

$$CVaR_\alpha^{F_Y^L} \leq CVaR_\alpha^{F_X} \leq CVaR_\alpha^{F_Y^U}, \tag{41}$$

where $CVaR_\alpha^{F_Y^L}, CVaR_\alpha^{F_X}$, and $CVaR_\alpha^{F_Y^U}$ denote the CVaR at level $\alpha$ corresponding to the distributions $F_Y^L$, $F_X$, and $F_Y^U$, respectively.

Let $a_{min} = min(a_X, a_Y)$ and $b_{min} = min(b_X, b_Y)$. Define the interval $[b_{min}, b_X)$ to be $\emptyset$ if $b_{min} = b_X$, and equal to $[b_Y, b_X)$ otherwise. Analogously, define $[a_{min}, a_X)$ in the same manner. We then define upper and lower bounds for $F_Y$ as follows (see Figure 1a):

$$F_Y^U(y) = \begin{cases} 0 & y < max(a_X, q_{1-\epsilon}^Y) \\ min(F_Y(y) - \epsilon, 1 - \epsilon) & y \in [max(a_X, q_{1-\epsilon}^Y), b_{min}) \\ 1 - \epsilon & y \in [b_{min}, b_X) \\ 1 & y \geq max(b_X, b_Y), \end{cases} \tag{42}$$

$$F_Y^L(y) = \begin{cases} 0 & y < a_{min} \\ \epsilon & y \in [a_{min}, a_Y) \\ min(F_Y(y) + \epsilon, 1) & y \in [a_Y, min(q_\epsilon^Y, b_X)) \\ 1 & y \geq min(q_\epsilon^Y, b_X). \end{cases} \tag{43}$$

As a first step, we verify that $F_Y^L$ and $F_Y^U$ are valid CDFs and that they satisfy $F_Y^L \leq F_X \leq F_Y^U$. To establish that $F$ is a CDF, it suffices to verify the following properties:

1. $F$ is non-decreasing;
2. $F : \mathbb{R} \to [0, 1]$ with $\lim_{x \to \infty} F(x) = 1$ and $\lim_{x \to -\infty} F(x) = 0$;
3. $F$ is right-continuous.

**Proof that $F_Y^L$ is a CDF:**

1. On the interval $[a_Y, min(q_{1-\epsilon}^Y, b_X))$, the function $F_Y^L$ is monotone increasing, since $F_Y$ is monotone increasing by virtue of being a CDF. Outside this interval, $F_Y^L$ is constant and consequently preserves monotonicity.

2. By definition, $\lim_{y \to \infty} F_Y^L(y) = 1$ and $\lim_{y \to -\infty} F_Y^L(y) = 0$. We need to show that $F_Y^L$ is bounded between 0 and 1. By its definition, $F_Y^L$ is bounded between 0 and 1.

3. Within the interval $[a_Y, min(q_\epsilon^Y, b_X))$, $F_Y^L$ is right-continuous since $F_Y$ is right-continuous as a CDF. Outside this interval, $F_Y^L$ is constant and hence also right-continuous.

**Proof that $F_Y^U$ is a CDF:**

1. On the interval $[max(a_X, q_{1-\epsilon}^Y), b_{min})$, the function $F_Y^U$ is monotone increasing, since $F_Y$ is monotonically increasing by virtue of being a CDF. Outside this interval, $F_Y^U$ is constant and consequently preserves monotonicity.

2. By definition, $\lim_{y \to \infty} F_Y^U(y) = 1$ and $\lim_{y \to -\infty} F_Y^U(y) = 0$. By its definition, $F_Y^U$ is bounded between 0 and 1.

3. Within the interval $[max(a_X, q_{1-\epsilon}^Y), b_{min})$, $F_Y^U$ is right-continuous since $F_Y$ is right-continuous as a CDF. Outside this interval, $F_Y^U$ is constant and hence also right-continuous.

**Proof that $F_Y^L \leq F_X \leq F_Y^U$:** To establish that $F_Y^L \leq F_X \leq F_Y^U$, we need to show that for all $z \in \mathbb{R}$, $F_Y^L(z) \geq F_X(z) \geq F_Y^U(z)$. Let $z \in \mathbb{R}$.

- If $z < a_{min}$ then $F_Y^L(z) = 0 = F_X(z)$.
- If $z \in [a_{min}, a_Y)$ then $F_X(y) = F_X(y) - F_Y(y) + F_Y(y) \leq |F_X(y) - F_Y(y)| + F_Y(y) \leq \epsilon + F_Y(y) = \epsilon = F_Y^L(y)$.
- If $z \in [a_Y, min(q_\epsilon^Y, b_X))$ we assume that $F_Y(z) + \epsilon \leq 1$ because otherwise $F_Y^L(z) = 1$ and the inequality holds. $F_Y^L(z) = F_Y(z) + \epsilon = F_Y(z) - F_X(z) + F_X(z) + \epsilon \geq F_X(z) - |F_Y(z) - F_X(z)| + \epsilon \geq F_X(z)$.
- If $z \geq min(q_\epsilon^Y, b_X)$ then $F_Y^L(z) = 1 \geq F_X(z)$.

and therefore $F_Y^L \leq F_X$. Note that the last proof holds when $b_X = \infty$, making $F_Y^L$ a valid CDF when the support of $X$ is not bounded from above.

- If $z < max(a_X, q_{1-\epsilon}^Y)$ then $F_Y^U(z) = 0 \leq F_X(z)$.
- If $z \in [max(a_X, q_{1-\epsilon}^Y), b_{min})$ then $F_Y^U(z) \leq F_Y(z) - \epsilon = F_Y(z) - F_X(z) + F_X(z) - \epsilon \leq |F_Y(z) - F_X(z)| + F_X(z) - \epsilon \leq \epsilon + F_X(z) - \epsilon = F_X(z)$.
- If $z \in [b_{min}, b_X)$ then $F_Y^U(z) = 1 - \epsilon = F_Y(z) - \epsilon = F_Y(z) - \epsilon + F_X(z) - F_X(z) \leq F_X(z) - \epsilon + |F_Y(z) - F_X(z)| \leq F_X(z)$

and therefore $F_X \leq F_Y^U$. Note that the last proof holds when $a_X = -\infty$, making $F_Y^U$ a valid CDF when the support of $X$ is not bounded from below.

Next, we derive bounds for the CVaR associated with $F_Y^U$ and $F_Y^L$.

**Upper bound for $CVaR_\alpha^{F_Y^U}$:** From Acerbi & Tasche (2002), CVaR is equal to an integral of the VaR

$$CVaR_\alpha^{F_Y^U} = \frac{1}{\alpha} \int_{1-\alpha}^{1} \inf\{y \in \mathbb{R} : F_Y^U(y) \geq \tau\} d\tau = \frac{1}{\alpha} \int_{1-\alpha+\epsilon}^{1+\epsilon} \inf\{y \in \mathbb{R} : F_Y^U(y) \geq \tau - \epsilon\} d\tau$$

(44)

If $\alpha \leq \epsilon$, then $1 - \alpha + \epsilon \geq 1$, rendering the bound in the preceding equation trivial, as it attains the maximum value of the support of both $X$ and $Y$.

$$\frac{1}{\alpha} \int_{1-\alpha+\epsilon}^{1+\epsilon} \inf\{y \in \mathbb{R} : F_Y^U(y) \geq \tau - \epsilon\} \leq \frac{1}{\alpha} \int_{1-\alpha+\epsilon}^{1+\epsilon} max(b_X, b_Y) d\tau = max(b_X, b_Y). \tag{45}$$

That is, $CVaR_\alpha(X) \leq max(b_X, b_Y)$.

If $\alpha > \epsilon$, then $1 - \alpha + \epsilon < 0$, and the integral may be decomposed into one term that is trivially bounded by the maximum of the support and another term that can be computed explicitly

$$CVaR_\alpha^{F_Y^U} = \frac{1}{\alpha} \int_{1-\alpha+\epsilon}^{1+\epsilon} \inf\{y \in \mathbb{R} : F_Y^U(y) \geq \tau - \epsilon\} d\tau$$

$$= \frac{1}{\alpha} [\underbrace{\int_1^{1+\epsilon} \inf\{y \in \mathbb{R} : F_Y^U(y) \geq \tau - \epsilon\} d\tau}_{\triangleq A_1} + \underbrace{\int_{1-\alpha+\epsilon}^1 \inf\{y \in \mathbb{R} : F_Y^U(y) \geq \tau - \epsilon\} d\tau}_{\triangleq A_2}] \tag{46}$$

The term $A_1$ is bounded, in a manner analogous to equation 45, by $\epsilon \max(b_X, b_Y)$, which is essentially the tightest bound attainable given the definition of $F_Y^U$. The term $A_2$ can be expressed in terms of the CVaR of $Y$, evaluated at a shifted confidence level. Specifically, the variable $\tau - \epsilon$ in $A_2$ lies within the interval $[1 - \alpha, 1 - \epsilon]$. Over this range, for all $y \in [\max(a_X, q_{1-\epsilon}^Y), b_{\min}]$, the inequality $F_Y^U(y) \leq F_Y(y) - \epsilon$ holds. This follows because if $q_{1-\epsilon}^Y \geq a_X$, then by definition $F_Y^U(y) = min(F_Y(y) - \epsilon, 1 - \epsilon)$, and if $a_X > q_{1-\epsilon}^Y$, then $F_Y^U(y) = 0 \leq F_Y(y) - \epsilon$ for $y \in [q_{1-\epsilon}^Y, a_X]$ and again $F_Y^U(y) = min(F_Y(y) - \epsilon, 1 - \epsilon)$ for $y \in [a_X, b_{min}]$.

$$A_2 \leq \int_{1-\alpha+\epsilon}^1 \inf\{y \in \mathbb{R} : F_Y(y) - \epsilon \geq \tau - \epsilon\} d\tau = \int_{1-(\alpha-\epsilon)}^1 \inf\{y \in \mathbb{R} : F_Y(y) \geq \tau\} d\tau$$

$$= (\alpha - \epsilon) CVaR_{\alpha-\epsilon}(Y). \tag{47}$$

Note that the confidence level is valid, as $\alpha, \epsilon \in (0, 1)$ and $\alpha > \epsilon$ imply $\alpha - \epsilon \in (0, 1)$. By combining the previous two expressions, we obtain a single bound:

$$CVaR_\alpha^{F_Y^U} \leq \epsilon \frac{max(b_X, b_Y)}{\alpha} + (1 - \frac{\epsilon}{\alpha}) CVaR_{\epsilon-\alpha}(Y). \tag{48}$$

In the case where $a_X = -\infty$, we have $\max(a_X, q_{1-\epsilon}^Y) = q_{1-\epsilon}^Y$, and the definition of $F_Y^U$ no longer involves $a_X$. Therefore, the inequality remains valid in the case of $a_X = -\infty$.

**Lower bound for** $CVaR_\alpha^{F_Y^L}$**:** From Acerbi & Tasche (2002), CVaR is equal to an integral of the VaR

$$CVaR_\alpha^{F_Y^L} = \frac{1}{\alpha} \int_{1-\alpha}^1 \inf\{y \in \mathbb{R} : F_Y^L(y) \geq \tau\} d\tau = \frac{1}{\alpha} \int_{1-(\epsilon+\alpha)}^{1-\epsilon} \inf\{y \in \mathbb{R} : F_Y^L(y) \geq \tau + \epsilon\} d\tau. \tag{49}$$

It holds that $F_Y^L(y) \leq F_Y(y) + \epsilon$ and therefore

$$CVaR_\alpha^{F_Y^L} \geq \frac{1}{\alpha} \int_{1-(\epsilon+\alpha)}^{1-\epsilon} \inf\{y \in \mathbb{R} : F_Y(y) + \epsilon \geq \tau + \epsilon\} d\tau = \frac{1}{\alpha} \int_{1-(\epsilon+\alpha)}^{1-\epsilon} \inf\{y \in \mathbb{R} : F_Y(y) \geq \tau\} d\tau$$

$$= \frac{1}{\alpha} [\int_{1-(\epsilon+\alpha)}^1 \inf\{y \in \mathbb{R} : F_Y(y) \geq \tau\} d\tau - \int_{1-\epsilon}^1 \inf\{y \in \mathbb{R} : F_Y(y) \geq \tau\} d\tau]$$

$$= \frac{1}{\alpha} [(\alpha + \epsilon) CVaR_{\alpha+\epsilon}(Y) - \epsilon CVaR_\epsilon(Y)]. \tag{50}$$

In the case where $\epsilon + \alpha > 1$,

$$CVaR_\alpha^{F_Y^L} = \frac{1}{\alpha} \int_{1-(\epsilon+\alpha)}^{1-\epsilon} \inf\{y \in \mathbb{R} : F_Y^L(y) \geq \tau + \epsilon\} d\tau$$

$$= \frac{1}{\alpha} [\underbrace{\int_0^{1-\epsilon} \inf\{y \in \mathbb{R} : F_Y^L(y) \geq \tau + \epsilon\} d\tau}_{\triangleq A_3} + \underbrace{\int_{1-(\epsilon+\alpha)}^0 \inf\{y \in \mathbb{R} : F_Y^L(y) \geq \tau + \epsilon\} d\tau}_{\triangleq A_4}]. \tag{51}$$

$$A_3 \geq \int_0^{1-\epsilon} \inf\{y \in \mathbb{R} : F_Y(y) \geq \tau\}d\tau = \int_0^1 \inf\{y \in \mathbb{R} : F_Y(y) \geq \tau\}d\tau$$

$$- \int_{1-\epsilon}^1 \inf\{y \in \mathbb{R} : F_Y(y) \geq \tau\}d\tau = \mathbb{E}[Y] - \epsilon CVaR_\epsilon(Y) \tag{52}$$

In the case of $A_4$, $\tau$ ranges from $1 - \alpha$ to $\epsilon$, and therefore

$$A_4 = \int_{1-\alpha}^\epsilon \inf\{y \in \mathbb{R} : F_Y^L(y) \geq \tau\}d\tau \geq (\alpha + \epsilon - 1)a_{min} \tag{53}$$

By combining the last equations to one bound we get

$$CVaR_\alpha(X) \geq CVaR_\alpha^{F_Y^L} \geq \mathbb{E}[Y] - \epsilon CVaR_\epsilon(Y) + (\alpha + \epsilon - 1)a_{min}. \tag{54}$$

$\square$

**Theorem B.2** *(Theorem 4.2) Given the definition of $X, Y, \epsilon$ and $\alpha$ as in Theorem 4.1, the lower and upper bounds of Theorem 4.1 converge to $CVaR_\alpha(X)$ as $\epsilon \to 0$.*

*Proof.* Let $f$ and $g$ be continuous functions such that $f(0)$ and $g(0)$ are finite. Then $\lim_{x\to 0} f(x)g(x) = \big(\lim_{x\to 0} f(x)\big)\big(\lim_{x\to 0} g(x)\big)$. This property will be used throughout the proof.

Assume that $F_Y(z) - F_X(z) \leq \epsilon$ for all $z \in \mathbb{R}$. Since we consider the limit as $\epsilon \to 0$, we further assume that $\alpha > \epsilon$ when computing the bound. Noting that CVaR is continuous with respect to the confidence level, it follows that $\lim_{\epsilon\to 0} \text{CVaR}_{\alpha-\epsilon}(Y) = \text{CVaR}_\alpha(Y)$.

$$\lim_{\epsilon\to 0} \frac{\epsilon}{\alpha} max(b_X, b_Y) + (1 - \frac{\epsilon}{\alpha})CVaR_{\alpha-\epsilon}(Y) = \lim_{\epsilon\to 0}(1 - \frac{\epsilon}{\alpha})CVaR_{\alpha-\epsilon}(Y)$$

$$= \lim_{\epsilon\to 0} 1 - \frac{\epsilon}{\alpha} \lim_{\epsilon\to 0} CVaR_{\alpha-\epsilon}(Y) = CVaR_\alpha(Y) \tag{55}$$

If $F_X(z) - F_Y(z) \leq \epsilon$ for all $z \in \mathbb{R}$, then, since we consider the limit as $\epsilon \to 0$, we assume in the derivation of the bound that $\alpha + \epsilon \leq 1$.

$$\lim_{\epsilon\to 0}(1 + \frac{\epsilon}{\alpha})CVaR_{\alpha+\epsilon}(Y) - \frac{\epsilon}{\alpha}CVaR_\epsilon(Y) = \lim_{\epsilon\to 0} 1 + \frac{\epsilon}{\alpha} \lim_{\epsilon\to 0} CVaR_{\alpha+\epsilon}(Y)$$

$$- \lim_{\epsilon\to 0} \frac{\epsilon}{\alpha} \lim_{\epsilon\to 0} CVaR_{\alpha+\epsilon}(Y) = CVaR_\alpha(Y). \tag{56}$$

$\square$

**Theorem B.3** *(Theorem 4.3: Tighter CVaR Lower Bound) Let $\alpha \in (0, 1)$, $X$ and $Y$ be random variables. Define the a random variable $Y^L$ such that $F_{Y^L}(y) \triangleq min(1, F_Y(y) + g(y))$ for $g : \mathbb{R} \to [0, \infty)$. Assume $\lim_{x\to-\infty} g(x) = 0$, $g$ is continuous from the right and monotonic increasing. If $\forall x \in \mathbb{R}, F_X(x) \leq F_Y(x) + g(x)$, then $F_{Y^L}$ is a CDF and $CVaR_\alpha(Y^L) \leq CVaR_\alpha(X)$.*

*Proof.* In order to prove that $F_{Y^L}$ is a CDF we need to prove that $F_{Y^L}$ is:

1. Monotonic increasing

2. $F : \mathbb{R} :\to [0, 1], \lim_{x\to\infty} F_{Y^L}(x) = 1, \lim_{x\to-\infty} F_{Y^L}(x) = 0$

3. Continuous from the right

**Monotonic increasing:** Note that for every $f_i : \mathbb{R} \to \mathbb{R}, i = 1, 2$ that are monotonic increasing, $f_1(f_2(x)))$ is also monotonic increasing in x. Denote $f(x) := min(x, 1)$ and $f_2(x) := F_Y(x) + g(x)$. $F_Y(x)$ is a CDF and therefore monotonic increasing, so $f_2(x)$ is monotonic increasing as a sum of monotonic increasing functions. $f_1$ is also monotonic increasing, and $F_{Y^L}(x) = f_1(f_2(x))$. Therefore $F_{Y^L}(x)$ is monotonic increasing.

**Limits:**

$$1 \geq \lim_{x \to \infty} F_{Y^L}(x) = \lim_{x \to \infty} min(1, F_Y(x) + g(x))$$
$$\geq \lim_{x \to \infty} min(1, F_Y(x)) = \lim_{x \to \infty} F_Y(x) = 1 \tag{57}$$

and therefore $\lim_{x \to \infty} F_{Y^L}(x) = 1$.

$$0 \leq \lim_{x \to -\infty} F_{Y^L}(x) = \lim_{x \to -\infty} min(1, F_Y(x) + g(x))$$
$$\leq \lim_{x \to -\infty} F_Y(x) + g(x) = \lim_{x \to -\infty} F_Y(x) + \lim_{x \to -\infty} g(x)$$
$$= 0$$

and therefore $\lim_{x \to -\infty} F_{Y^L}(x) = 0$. By definition $\forall x \in \mathbb{R}, F_{Y^L}(x) \leq 1$, and $\forall x \in \mathbb{R}, F_{Y^L}(x) \geq 0$ because both g and $F_{Y^L}$ are non negative functions.

**Continuity from the right:** $F_Y$ is continuous from the right because it is a CDF, and therefore $F_{Y^L}$ is continuous from the right as a sum of continuous from the right functions.
Thus, $F_{Y^L}$ is a CDF.

**Bound proof:** If $Y^L \leq X$, then $CVaR_\alpha(Y^L) \leq CVaR_\alpha(X)$ because CVaR is a coherent risk measure. Note that if $F_Y(x) + g(x) < 1$ then

$$F_X(x) \leq F_Y(x) + g(x) = F_{Y^L}(x), \tag{58}$$

and if $F_Y(x) + g(x) \geq 1, 1 = F_{Y^L}(x) \geq F_X(x)$. Therefore $Y^L \leq X$. $\qquad \square$

**Theorem B.4** *(Theorem 4.4) Let $\alpha \in (0, 1)$, $X$ and $Y$ random variables. Define $h : \mathbb{R} \to [0, \infty)$ to be a continuous function, $g(z) := \int_{-\infty}^{z} h(x)dx$ and $Y^L$ to be a random variable such that $F_{Y^L}(y) := \min(1, F_Y(y) + g(y))$. If $\lim_{z \to -\infty} g(z) = 0$ and $\forall x \in \mathbb{R}, f_x(x) \leq f_y(z) + h(x)$, then $F_{Y^L}$ is a CDF and $CVaR_\alpha(Y^L) \leq CVaR_\alpha(X)$.*

*Proof.* We will show the g satisfies the properties of Theorem 4.3, and therefore this theorem holds. We need to prove that

1. $\lim_{z \to -\infty} g(z) = 0$

2. g is continuous from the right.

3. g is monotonic increasing.

4. $F_X(y) \leq F_Y(y) + g(y)$

It is given in the theorem's assumptions that $\lim_{z \to -\infty} g(z) = 0$, so (1) holds. h is non negative and therefore $g$ is monotonic increasing, so (3) holds. $g$ is continuous if its derivative exists for all $z \in \mathbb{R}$. Let $z \in \mathbb{R}$ and $a < z$.

$$\frac{d}{dz}g(z) = \frac{d}{dz}\int_{-\infty}^{z} h(x)dx = \frac{d}{dz}[\int_{-\infty}^{a} h(x)dx + \int_{a}^{z} h(x)dx] = \frac{d}{dz}\int_{a}^{z} h(x)dx = h(z) \tag{59}$$

where the third equality holds because $\int_{-\infty}^{a} h(x)dx = g(a)$ is a constant that does not depend on z. The last equality holds from the fundamental theorem of calculus because $h$ is continuous. Finally, (4) holds because

$$F_X(y) \triangleq \int_{-\infty}^{y} f_x(x)dx \leq \int_{-\infty}^{y} f_y(x) + h(x)dx \triangleq F_Y(y) + g(y). \tag{60}$$

$$\qquad \square$$

## C  PROOFS OF SECTION 5

**Theorem C.1** *(Theorem 5.1) Let $X$ a random variable, $\alpha \in (0, 1], \delta \in (0, 1), \epsilon = \sqrt{\ln(1/\delta)/(2n)}$. Let $X_1, \ldots, X_n \overset{iid}{\sim} F_X$ be random variables that define the ECDF $\hat{F}_X$.*

1. **Upper Bound:** If $P(X \le b) = 1$, then

   (a) If $\alpha > \epsilon$ then $P(CVaR_\alpha(X) \le (1 - \frac{\epsilon}{\alpha})C_{\alpha-\epsilon}^{\hat{F}_X} + \frac{\epsilon}{\alpha}b) > 1 - \delta$.

   (b) If $\alpha \le \epsilon$ then $CVaR_\alpha(X) \le b$

2. **Lower Bound:** If $P(X \ge a) = 1$, then

   (a) If $\alpha + \epsilon < 1$, then $P(CVaR_\alpha(X) \ge (1 + \frac{\epsilon}{\alpha})C_{\alpha+\epsilon}^{\hat{F}_X} - \frac{\epsilon}{\alpha}C_\epsilon^{\hat{F}_X}) > 1 - \delta$.

   (b) If $\alpha + \epsilon \ge 1$, then $P(CVaR_\alpha(X) \ge \frac{1}{\alpha}[(\alpha + \epsilon - 1)a + E_{\hat{F}_X}[X] - \epsilon C_\epsilon^{\hat{F}_X}]) > 1 - \delta$.

*Proof.* From DKW inequality the following inequalities can be derived Thomas & Learned-Miller (2019)

$$\Pr\left(\sup_{x\in\mathbb{R}}\left(\hat{F}(x) - F(x)\right) \le \sqrt{\frac{\ln(1/\delta)}{2n}}\right) \ge 1 - \delta, \tag{61}$$

$$\Pr\left(\sup_{x\in\mathbb{R}}\left(\hat{F}(x) - F(x)\right) \ge \sqrt{\frac{\ln(1/\delta)}{2n}}\right) \ge 1 - \delta. \tag{62}$$

Let $\epsilon = \sqrt{\frac{\ln(1/\delta)}{2n}}$. By using Theorem 4.1 we get the following.

If $\alpha < \epsilon$ then

$$P(CVaR_\alpha(X) \le \frac{\alpha - \epsilon}{\alpha}C_{\alpha-\epsilon}^{\hat{F}_X} + \frac{\epsilon}{\alpha}b)$$

$$= \underbrace{P(C_\alpha(X) \le \frac{\alpha - \epsilon}{\alpha}C_{\alpha-\epsilon}^{\hat{F}_X} + \frac{\epsilon}{\alpha}b \Big| \sup_{z\in\mathbb{R}}(\hat{F}_X - F_X) \le \epsilon)}_{=1} \times \underbrace{P(\sup_{z\in\mathbb{R}}(\hat{F}_X - F_X) \le \epsilon)}_{>1-\delta}$$

$$+ \underbrace{P(C_\alpha(X) \le \frac{\alpha - \epsilon}{\alpha}C_{\alpha-\epsilon}^{\hat{F}_X} + \frac{\epsilon}{\alpha}b \Big| \sup_{z\in\mathbb{R}}(\hat{F}_X - F_X) > \epsilon)}_{\ge 0} \times \underbrace{P(\sup_{z\in\mathbb{R}}(\hat{F}_X - F_X) > \epsilon)}_{\ge 0} \tag{63}$$

$$> 1 - \delta.$$

Observe that, conditional on the event $\sup_{z\in\mathbb{R}}\left(\hat{F}_X(z) - F_X(z)\right) \le \epsilon$, the bound in Theorem 4.1 holds deterministically. Consequently, the probability that

$$C_\alpha(X) \le \frac{\alpha - \epsilon}{\alpha}C_{\alpha-\epsilon}^{\hat{F}_X} + \frac{\epsilon}{\alpha}b \tag{64}$$

holds, given $\sup_{z\in\mathbb{R}}\left(\hat{F}_X(z) - F_X(z)\right) \le \epsilon$, is equal to one. The same observation is necessary through the rest of the proof in a similar manner. If $\alpha + \epsilon < 1$, then

$$P(C_\alpha(X) \ge (1 + \frac{\epsilon}{\alpha})C_{\alpha+\epsilon}^{\hat{F}_X} - \frac{\epsilon}{\alpha}C_\epsilon^{\hat{F}_X})$$

$$= \underbrace{P(C_\alpha(X) \ge (1 + \frac{\epsilon}{\alpha})C_{\alpha+\epsilon}^{\hat{F}_X} - \frac{\epsilon}{\alpha}C_\epsilon^{\hat{F}_X} \Big| \sup_{z\in\mathbb{R}}(F_X - \hat{F}_X) \le \epsilon)}_{=1} \underbrace{P(\sup_{z\in\mathbb{R}}(F_X - \hat{F}_X) \le \epsilon)}_{>1-\delta}$$

$$+ \underbrace{P(C_\alpha(X) \ge (1 + \frac{\epsilon}{\alpha})C_{\alpha+\epsilon}^{\hat{F}_X} - \frac{\epsilon}{\alpha}C_\epsilon^{\hat{F}_X} \Big| \sup_{z\in\mathbb{R}}(F_X - \hat{F}_X) > \epsilon)}_{\ge 0} \times \underbrace{P(\sup_{z\in\mathbb{R}}(F_X - \hat{F}_X) > \epsilon)}_{\ge 0} \tag{65}$$

$$> 1 - \delta$$

If $\alpha + \epsilon \geq 1$, then

$$P(C_\alpha(X) \geq \frac{1}{\alpha}[(\alpha + \epsilon - 1)a + E_{\hat{F}_X}[X] - \epsilon C_\epsilon^{\hat{F}_X}])$$

$$= \underbrace{P(C_\alpha(X) \geq \frac{1}{\alpha}[(\alpha + \epsilon - 1)a + E_{\hat{F}_X}[X] - \epsilon C_\epsilon^{\hat{F}_X}] \Big| \sup_{z \in \mathbb{R}}(F_X - \hat{F}_X) \leq \epsilon)}_{=1} \times \underbrace{P(\sup_{z \in \mathbb{R}}(F_X - \hat{F}_X) \leq \epsilon)}_{> 1 - \delta}$$

$$+ \underbrace{P(C_\alpha(X) \geq \frac{1}{\alpha}[(\alpha + \epsilon - 1)a + E_{\hat{F}_X}[X] - \epsilon C_\epsilon^{\hat{F}_X}] \Big| \sup_{z \in \mathbb{R}}(F_X - \hat{F}_X) > \epsilon)}_{\geq 0} \times \underbrace{P(\sup_{z \in \mathbb{R}}(F_X - \hat{F}_X) > \epsilon)}_{\geq 0}$$

$$> 1 - \delta$$

$$\tag{66}$$

$\square$

**Corollary C.2** *(Corollary 5.2) Let $X$ be a random variable, $\alpha \in (0,1], \delta \in (0,1), \epsilon = \sqrt{\ln(1/\delta)/(2n)}, a \in \mathbb{R}, b \in \mathbb{R}, \eta > 0$. Let $X_1, \ldots, X_n \overset{iid}{\sim} F_X$ be random variables that define the ECDF $\hat{F}_X$. Denote by $U(n)$ and $L(n)$ the upper and lower bounds respectively from Theorem 5.1, where $n$ is the number of samples, then*

1. *If $P(X \leq b) = 1$, then $\lim_{n \to \infty} U(n) \overset{a.s}{=} CVaR_\alpha(X)$.*

2. *If $P(X \geq a) = 1$, then $\lim_{n \to \infty} L(n) \overset{a.s}{=} CVaR_\alpha(X)$.*

*where a.s denotes almost sure convergence.*

*Proof.*

$$\lim_{n \to \infty} U(n) \overset{a.s}{=} \lim_{n \to \infty} (1 - \frac{\epsilon}{\alpha})C_{\alpha - \epsilon}^{\hat{F}_X} + \frac{\epsilon}{\alpha}b \overset{a.s}{=} \lim_{n \to \infty} C_{\alpha - \epsilon}^{\hat{F}_X} = \lim_{n \to \infty} C_\alpha^{\hat{F}_X} = C_\alpha^{F_X} \tag{67}$$

where the first and second equalities follow from the fact that $\epsilon \to 0$ as $n \to \infty$; the third equality holds by the continuity of CVaR with respect to $\alpha$; and the fourth equality holds by the almost sure convergence of the empirical CVaR of i.i.d. samples to the true CVaR. For the same reason,

$$\lim_{n \to \infty} L(n) \overset{a.s}{=} \lim_{n \to \infty} (1 + \frac{\epsilon}{\alpha})C_{\alpha + \epsilon}^{\hat{F}_X} - \frac{\epsilon}{\alpha}C_\epsilon^{\hat{F}_X} \overset{a.s}{=} \lim_{n \to \infty} C_{\alpha + \epsilon}^{\hat{F}_X} \overset{a.s}{=} \lim_{n \to \infty} C_\alpha^{\hat{F}_X} \overset{a.s}{=} \lim_{n \to \infty} C_\alpha^{F_X}. \tag{68}$$

$\square$

**Theorem C.3** *(Theorem 5.3) Let $X$ and $Y$ be random variables, $\epsilon \in [0,1]$, and $\eta = \sqrt{\ln(1/\delta)/(2n)}, \epsilon' = min(\epsilon + \eta, 1)$. Let $Y_1, \ldots, Y_n$ be independent and identically distributed samples from $F_Y$, and denote by $\hat{F}_Y$ the associated empirical cumulative distribution function.*

1. ***Upper Bound:*** *If $\forall z \in \mathbb{R}, F_Y(z) - F_X(z) \leq \epsilon$ and $P(X \leq b_X) = 1, P(Y \leq b_Y) = 1$, then*

   *(a) If $\alpha > \epsilon'$ then*

   $$P\left(CVaR_\alpha(X) \leq \frac{\epsilon'}{\alpha}max(b_X, b_Y) + (1 - \frac{\epsilon'}{\alpha})CVaR_{\alpha - \epsilon'}^{\hat{F}_Y^L}\right) > 1 - \delta. \tag{69}$$

   *(b) If $\alpha \leq \epsilon'$, then $CVaR_\alpha(X) \leq max(b_X, b_Y)$*

2. ***Lower Bound:*** *If $\forall z \in \mathbb{R}, F_X(z) - F_Y(z) \leq \epsilon$ and $P(X \geq a_X) = 1, P(Y \geq a_Y) = 1$, then*

   *(a) If $\alpha + \epsilon' \leq 1$, then*

   $$P\left(CVaR_\alpha(X) \geq (1 + \frac{\epsilon'}{\alpha})CVaR_{\alpha + \epsilon'}^{\hat{F}_Y^L} - \frac{\epsilon'}{\alpha}CVaR_{\epsilon'}^{\hat{F}_Y^L}\right) > 1 - \delta. \tag{70}$$

(b) If $\alpha + \epsilon' > 1$, then

$$P\left(CVaR_\alpha(X) \geq \mathbb{E}_{\hat{F}_Y^L}[Y] - \epsilon' CVaR_{\epsilon'}^{\hat{F}_Y} + (\alpha + \epsilon' - 1)a_{min}\right) > 1 - \delta. \quad (71)$$

*Proof.* We begin by establishing that $\epsilon'$ bounds the distributional discrepancy between $\hat{F}_Y^L$ and $F_X$, under the assumption that $\sup_x \left(\hat{F}_Y^L(x) - F_Y(x)\right) \leq \epsilon$. Let $x \in \mathbb{R}$,

$$\hat{F}_Y^L(x) - F_X(x) = \hat{F}_Y^L(x) - F_Y(x) + F_Y(x) - F_X(x)$$
$$\leq |\hat{F}_Y^L(x) - F_Y(x)| + |F_Y(x) - F_X(x)| \leq \epsilon + \eta = \epsilon'. \quad (72)$$

Assume that $\alpha > \epsilon'$. Note that, conditional on the event $\sup_{z \in \mathbb{R}} \left(\hat{F}_Y(z) - F_X(z)\right) \leq \epsilon'$, the upper bound in Theorem 4.1 holds deterministically. Consequently, the probability that

$$CVaR_\alpha(X) \leq \frac{\epsilon'}{\alpha}max(b_X, b_Y) + (1 - \frac{\epsilon'}{\alpha})CVaR_{\alpha-\epsilon'}^{\hat{F}_Y^L} \quad (73)$$

holds is equal to one. From the law of total probability,

$$P\left(CVaR_\alpha(X) \leq \frac{\epsilon'}{\alpha}max(b_X, b_Y) + (1 - \frac{\epsilon'}{\alpha})CVaR_{\alpha-\epsilon'}^{\hat{F}_Y^L}\right) \geq P(\sup_{z \in \mathbb{R}} \left(\hat{F}_Y(z) - F_X(z)\right) \leq \epsilon'). \quad (74)$$

The full derivation is available in the Appendix.

$$P\left(CVaR_\alpha(X) \leq \frac{\epsilon'}{\alpha}max(b_X, b_Y) + (1 - \frac{\epsilon'}{\alpha})CVaR_{\alpha-\epsilon'}^{\hat{F}_Y^L}\right)$$
$$= P\left(CVaR_\alpha(X) \leq \frac{\epsilon'}{\alpha}max(b_X, b_Y) + (1 - \frac{\epsilon'}{\alpha})CVaR_{\alpha-\epsilon'}^{\hat{F}_Y^L}\right.$$
$$\left.\Big| \sup_{z \in \mathbb{R}} \left(\hat{F}_Y(z) - F_X(z)\right) \leq \epsilon'\right)P(\sup_{z \in \mathbb{R}} \left(\hat{F}_Y(z) - F_X(z)\right) \leq \epsilon')$$
$$+ P\left(CVaR_\alpha(X) \leq \frac{\epsilon'}{\alpha}max(b_X, b_Y) + (1 - \frac{\epsilon'}{\alpha})CVaR_{\alpha-\epsilon'}^{\hat{F}_Y^L}\right.$$
$$\left.\Big| \sup_{z \in \mathbb{R}} \left(\hat{F}_Y(z) - F_X(z)\right) > \epsilon'\right)P(\sup_{z \in \mathbb{R}} \left(\hat{F}_Y(z) - F_X(z)\right) > \epsilon') \quad (75)$$
$$= P(\sup_{z \in \mathbb{R}} \left(\hat{F}_Y(z) - F_X(z)\right) \leq \epsilon')$$
$$+ P\left(CVaR_\alpha(X) \leq \frac{\epsilon'}{\alpha}max(b_X, b_Y) + (1 - \frac{\epsilon'}{\alpha})CVaR_{\alpha-\epsilon'}^{\hat{F}_Y^L}\right.$$
$$\left.\Big| \sup_{z \in \mathbb{R}} \left(\hat{F}_Y(z) - F_X(z)\right) > \epsilon'\right)P(\sup_{z \in \mathbb{R}} \left(\hat{F}_Y(z) - F_X(z)\right) > \epsilon')$$
$$\geq P(\sup_{z \in \mathbb{R}} \left(\hat{F}_Y(z) - F_X(z)\right) \leq \epsilon')$$

From DKW Dvoretzky et al. (1956) inequality the following inequalities can be derived Thomas & Learned-Miller (2019)

$$P\left(\sup_{x \in \mathbb{R}} \left(\hat{F}(x) - F(x)\right) \leq \sqrt{\frac{\ln(1/\delta)}{2n}}\right) \geq 1 - \delta, P\left(\sup_{x \in \mathbb{R}} \left(\hat{F}(x) - F(x)\right) \geq \sqrt{\frac{\ln(1/\delta)}{2n}}\right) \geq 1 - \delta. \quad (76)$$

$$P(\sup_{z \in \mathbb{R}} \left(\hat{F}_Y(z) - F_X(z)\right) \leq \epsilon') \geq P(\sup_{z \in \mathbb{R}} \left(\hat{F}_Y(z) - F_Y(z)\right) + \sup_{z \in \mathbb{R}} \left(F_Y(z) - F_X(z)\right) \leq \epsilon')$$
$$\geq P(\sup_{z \in \mathbb{R}} \left(\hat{F}_Y(z) - F_Y(z)\right) + \epsilon \leq \epsilon + \eta) = P(\sup_{z \in \mathbb{R}} \left(\hat{F}_Y(z) - F_Y(z)\right) \leq \eta) > 1 - \delta. \quad (77)$$

The first inequality follows from the triangle inequality; the second holds since $\epsilon$ bounds the distributional discrepancy between $X$ and $Y$; and the third follows from equation 76.

As with the preceding equations, all bounds in Theorem 4.1 hold deterministically for a given distributional discrepancy, where the discrepancy between $\hat{F}_Y$ and $F_X$ is $\epsilon'$. Consequently, the remaining probabilistic guarantees hold with probability at least $1 - \delta$. $\square$

# D   OUR CONCENTRATION BOUNDS VS THOMAS & LEARNED-MILLER (2019)

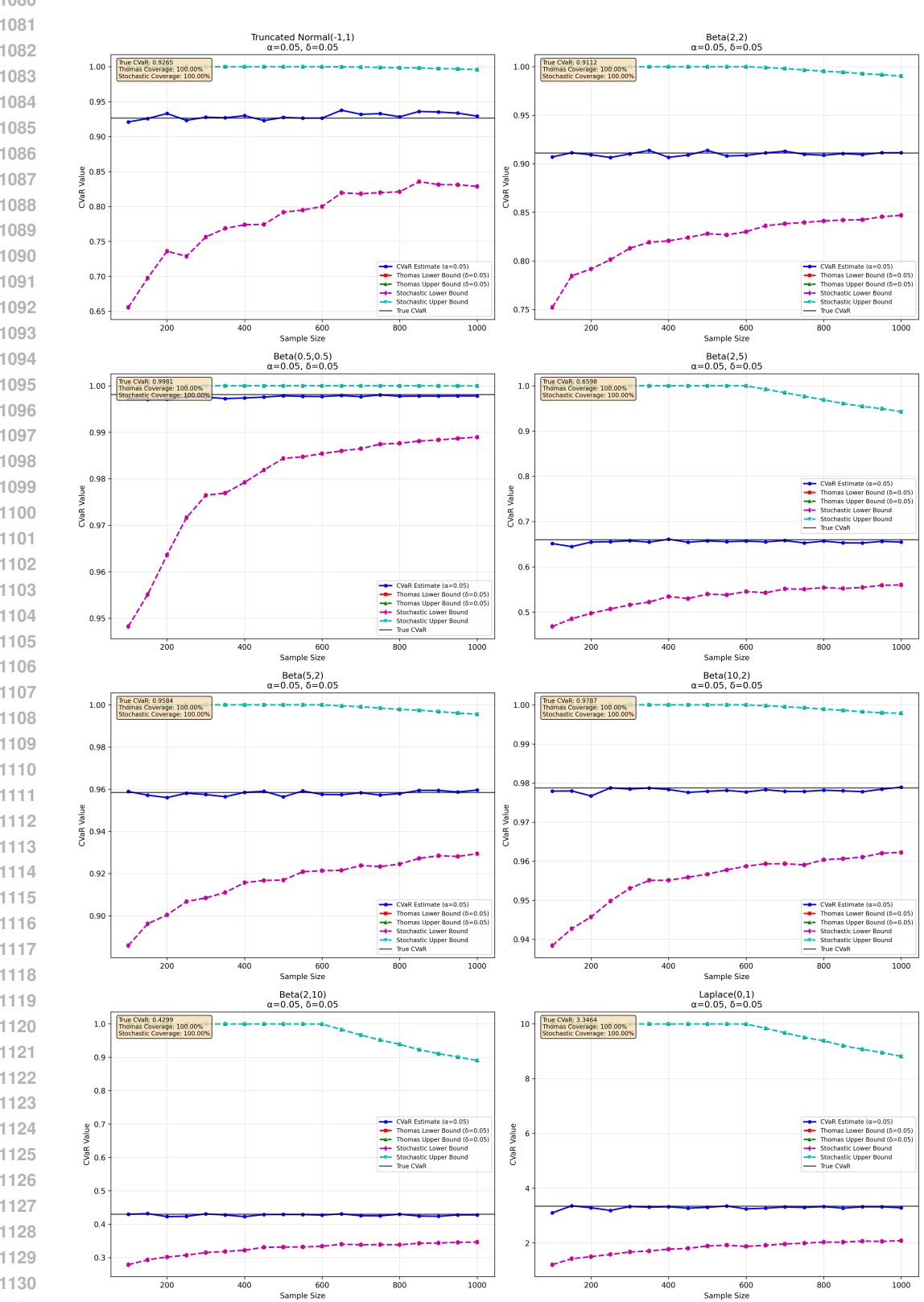

Figure 4: Comparison of concentration inequalities for CVaR.

