# OpenReview forum: "Bounding Conditional Value-at-Risk via Auxiliary Distributions with Bounded Discrepancies"
_ICLR.cc/2026/Conference — Submitted to ICLR 2026_

### Official Review · Reviewer_qxMt · 2025-10-28

**Soundness:** 3
**Presentation:** 3
**Contribution:** 2
**Rating:** 4
**Confidence:** 5

**Summary:**

The paper studies the Conditional Value at Risk which is a (popular) risk measure associated to a given random variable $X$ and has a wide range of applications. The paper consider the following problem. Supposed that two random variables $X$ and $Y$ are closed  in the sense that their corresponding CDF are uniformly closed to each other.  Can one  bound the conditional value at risk of $X$ by the conditional value of risk of $Y$?  The papers provides such bound and discuss also the special case of when $Y$ is is the emprical distribution associated to $n$ samples of $X$ in which case the bounds can be understood as concentration inequalities for  the conditional value at risk.

**Strengths:**

1) The paper is really very well written and the proofs are very well explained (and correct as far as I can judge).

2) The general problem of the sensitivity of the conditional value at risk, that is to understand how does the conditional value at risk chanae change when  distribution error (or only sampling are known) has broad interest in a variety of scientific fields.

**Weaknesses:**

1)  For random variables which are light-tailed (or bounded as considered in this paper), there are many coherent (or convex) risk measures  available.  The conditional value at risk is especially interesting for heavy-tailed random variables, such as power-law random variables because in that case other risk measures are then infinite.  The results in the paper deals with random variables which are bounded above or bounded below only but  provide then only upper bound or lower bound and do not really explore the tail behavior of the distribution.

2) The technique of proofs in the papers borrow very heavily (and streamline) previous results obtained by P. Thomas and E. Learned-Miller which dealt with concentration inequality.  The proof is essentially the same than in that paper and there is not too much technical novelty involved.

3) In the supplementary material an example from RL is discussed as a motivation but the results of the paper are not really applied in this context. The examples in the paper are a little too "textbook" style.

**Questions:**

1) Is there a sense in which the results of this paper are optimal, in the sense that the bounds are the worst possible. The random variables here are close in the  Kolmogorov-Smirnov metric. Is there a sense that the bounds are optimal among all random variables $Y$ which are say bounded and $\epsilon$-close in that metric.  Results in that direction for conditional value at risk exists, see for example the recent preprint by Anand Deo  https://arxiv.org/abs/2506.16230

---

### Official Review · Reviewer_2xzE · 2025-10-30

**Soundness:** 3
**Presentation:** 2
**Contribution:** 2
**Rating:** 4
**Confidence:** 4

**Summary:**

The paper focuses on the problem of estimating the conditional value at risk (CVAR) of a random variable given access to another random variable close to it. The conditional value at risk of a random variable $X$ at a given parameter $0 \leq \alpha \leq 1$ is the quantity $\mathbb{E} [X \mid X \geq q_{1 - \alpha}]$ where $q_{1 - \alpha}$ corresponds to the $1 - \alpha$-quantile of $X$. In this paper, the authors derive estimates of this quantity from another random variable whose cdf is close $X$. The authors provide upper and lower bounds on the CVAR of $X$ as a function of those of $Y$. Additionally, they derive finite sample guarantees which assume access to a small set of samples from $Y$.

My main concern with the paper is that it is unclear what the main contributions of the paper are. The proofs of the CDF-closeness to CVAR closeness results are quite standard and the extensions to finite-sample settings also rely on well-known concentration inequalities. Furthermore, the empirical utility of these bounds is also challenging to evaluate given the limited experiments in the paper. Finally, the motivation of the paper is also not clearly explained. It would be great if the authors could provide some clarification on this. It is not clear when access to a random variable of interest is limited but an easy to obtain surrogate is readily available and satisfies the pre-conditions of the theorems proved here. Clarifying a motivating application, expanding the empirical section, and improving the exposition of the technical contributions and their novelty would substantially strengthen the paper.

**Strengths:**

See main review

**Weaknesses:**

See main review

**Questions:**

See main review

---

### Official Review · Reviewer_BnYY · 2025-10-31

**Soundness:** 2
**Presentation:** 2
**Contribution:** 2
**Rating:** 2
**Confidence:** 4

**Summary:**

This paper develops a theoretical framework for bounding the Conditional Value-at-Risk (CVaR) of a random variable X (e.g., hard-to-sample or computationally expensive distributions) using an auxiliary random variable Y (more tractable/observable), under assumptions on their cumulative distribution functions (CDFs) and probability density functions (PDFs). CVaR is critical for risk-averse decision-making in AI (reinforcement learning, safe control) but often hard to estimate directly for complex X; the work addresses this by leveraging Y to derive interpretable, rigorous bounds.


Key Contributions.

1. The paper gives a general CVaR bounds. Theorem 4.1 establishes upper/lower bounds for \(CVaR_\alpha(X)\) via \(CVaR_{\alpha\pm\epsilon}(Y)\).

2. Theorems 5.1–5.3 extend bounds to empirical settings, using the Dvoretzky–Kiefer–Wolfowitz (DKW) inequality to control ECDF discrepancy. Notably, Thomas & Learned-Miller (2019)’s state-of-the-art CVaR concentration bounds emerge as a special case.

3. The framework accelerates risk-averse agents (e.g., POMDP planning), where X (original return distribution) and Y (simplified model) enable fast, guaranteed CVaR estimation for safe decision-making.

**Strengths:**

The paper demonstrates strong originality by addressing a critical gap in CVaR estimation: rigorous, interpretable bounds for hard-to-sample random variables X via tractable auxiliary Y—a need unmet by existing work. Unlike classical CVaR methods (relying on direct sampling of X) or Thomas & Learned-Miller (2019)’s concentration bounds (limited to empirical distributions and framed via order statistics), it innovatively constructs a framework linking \(CVaR_\alpha(X)\) to CVaR of Y using uniform/non-uniform CDF/PDF discrepancies. Therefore the problem setting is quite clear, and the motivation is good.

**Weaknesses:**

My main concern is the applicability of the results in the manuscript. See detailed comments in "Questions" section.

**Questions:**

1. To my understanding, CVaR is mostly used in the extreme value analysis, especially for heavy-tailed random variables. However, the current paper assumes the boundedness of the random variables. Could the authors extend the results to the unbounded and heavy-tailed setting. Otherwise, CVaR is not a very interesting statistic to be studied.

2. The numerical study given in the manuscript is not adequate.  Only the truncated mixture Gaussian random variable is considered. Could the authors give more challenging numerical examples to show the applicability of the proposed methods?

3. It would be better if the authors could highlight the technical challenges of deriving those concentration inequalities.

**Details Of Ethics Concerns:**

No concern.

---

### Official Review · Reviewer_1ERW · 2025-11-01

**Soundness:** 2
**Presentation:** 2
**Contribution:** 1
**Rating:** 2
**Confidence:** 4

**Summary:**

This paper develops a general theoretical framework for bounding the CVaR of a random variable X through another auxiliary variable Y, when their distributional discrepancy  is bounded.

**Strengths:**

- The theoretical derivations seem mostly sound and  consistent.
- The paper provides an alternative interpretable expression of existing concentration inequalities, replacing reweighted-order-statistic formulas by CVaR-form bounds.

**Weaknesses:**

- The novelty of theoretical results is very limited. The work seems to provide a unified and interpretable formulation of CVaR bounds and concentration inequalities, but the mathematical content closely mirrors existing literature. Most results restate or directly follow from existing CVaR inequalities (Brown 2007; Thomas & Learned-Miller 2019).  In addition, the claimed generalization to auxiliary distributions is conceptually interesting but limited in novelty. The key ideas of using stochastic dominance and DKW are well known in the literature. Thus, while the main results are correct, the contribution is quite incremental.
- The motivation is weak. The introduction does not clearly present a realistic scenario where Y is known and X is unknown yet related by a measurable discrepancy. The only sensible setting to me is when F_Y is the empirical distribution, but this has been widely studied.
- Some merely restate monotonicity of CVaR and should not appear as main results. Thm 4.3 and Thm 4.4 are direct results of monotonicity of CVaR.
- Lengthy derivations obscure the main message. Notation  occasionally inconsistent

**Questions:**

questions/comments
- Can the authors provide a concrete practical setting beyond the toy GMM example where X and Y are known to satisfy a computable discrepancy bound?
- Figures 1–2 help but the main motivation and use-cases are buried in the appendix instead of the introduction.

---

### Meta-Review · Area_Chair_4VaF · 2026-01-02

**Summary:**

This paper introduces bounds on the conditional value-at-risk (CVaR) of a computationally expensive random variable $X$ with the CVaR of a computationally tractable surrogate $Y$. While this paper is timely and well motivated in theory, it has several weaknesses that make it unsuitable for publication at ICLR:

* The authors do not provide sufficient real-world examples that justify the setting; i.e., a hard-to-sample real-world variable that admits a useful surrogate that satisfies the conditions of the theorem.
* The boundedness assumptions in the paper may limit its applicability, as CVaR is often used in extreme-value analysis.
* The proof techniques are borrowed heavily from the literature.
* The empirical analysis is limited.

The authors did not provide a rebuttal to address these weaknesses.

**Reviewer Concerns:**

N/A (no rebuttal)

**Reviewer Scores:**

N/A (no rebuttal)

---

### Decision · Program_Chairs · 2026-01-26

Reject